# Agonist efficiency links binding and gating in a nicotinic receptor

Dinesh C Indurthi, Anthony Auerbach*

Department of Physiology and Biophysics, University at Buffalo, State University of New York, Buffalo, United States

**Abstract** Receptors signal by switching between resting (C) and active (O) shapes ('gating') under the influence of agonists. The receptor's maximum response depends on the difference in agonist binding energy, O minus C. In nicotinic receptors, efficiency ($\eta$) represents the fraction of agonist binding energy applied to a local rearrangement (an induced fit) that initiates gating. In this receptor, free energy changes in gating and binding can be interchanged by the conversion factor $\eta$. Efficiencies estimated from concentration-response curves (23 agonists, 53 mutations) sort into five discrete classes (%): 0.56 (17), 0.51(32), 0.45(13), 0.41(26), and 0.31(12), implying that there are 5 C versus O binding site structural pairs. Within each class efficacy and affinity are corelated linearly, but multiple classes hide this relationship. $\eta$ unites agonist binding with receptor gating and calibrates one link in a chain of coupled domain rearrangements that comprises the allosteric transition of the protein.

## Editor's evaluation

This valuable work investigates the fundamental concept of how the energy of agonist binding is converted into the energy of the conformational change that opens the pore of a ligand-gated ion channel. The conclusions are based on analysis of solid data in terms of a mechanistic model. The findings will be interesting to biophysicists working on ligand-gated ion channels and, more generally, to enzymologists focused on allosteric enzyme regulation.

*For correspondence:
auerbach.anthony@gmail.com

**Competing interest:** The authors declare that no competing interests exist.

## Introduction

The primary job of a receptor is to convert chemical energy from ligand binding into mechanical work of protein conformational change. Each agonist has two affinities that measure how strongly it binds to resting and active target sites, and an efficacy that measures how well it activates once bound (*Cecchini and Changeux, 2022*; *Ehlert, 2015*; *Foreman et al., 2011*). A third agonist property, efficiency ($\eta$; eta), is the correlation between efficacy and affinity and is the receptor's output/input energy ratio (*Nayak et al., 2019*). Here, we describe and interpret the distribution of $\eta$ values estimated from concentration(dose)-response curves (CRCs) of adult-type skeletal muscle nicotinic acetylcholine receptors (AChRs) activated by different agonists and having different binding site mutations.

AChRs are five subunit, ligand-gated ion channels that have two neurotransmitter sites in the extracellular domain (ECD), and a narrow equatorial gate in the lumen of the transmembrane domain (TMD) that governs the passive flow of cations across the membrane (*Gharpure et al., 2020*; *Unwin, 1995*; *Zarkadas et al., 2022*). AChRs are allosteric proteins that alternate spontaneously between global off (C, closed-channel) and on (O, open-channel) conformations in which each neurotransmitter site binds agonists weakly (with low affinity; LA) or strongly (with high affinity; HA) (*Cecchini and Changeux, 2022*; *Jackson, 2006*; *Phillips, 2020*). The AChR neurotransmitter sites and gate are separated by ~60 Å, but both regions change structure and function within the gating isomerization, $C_{LA} \rightleftarrows O_{HA}$.

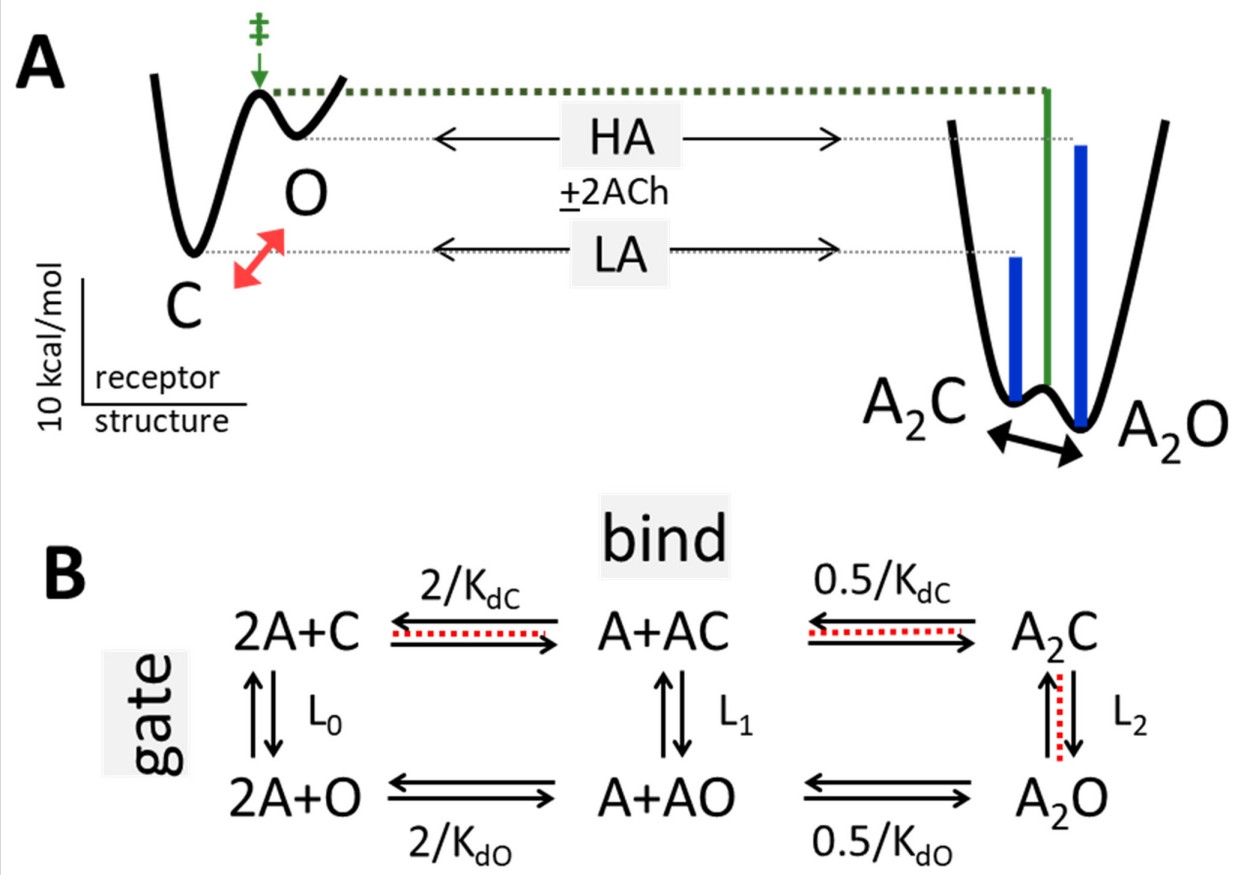

**Figure 1.** Bind and gate. (**A**) Potential energy surfaces (landscapes) for receptor activation without (left) and with (right) bound agonists (**A**). C, closed channel and low affinity (LA); O, open channel and high affinity (HA). Red, without agonists intrinsic C→O gating is uphill; blue lines, agonists increase $P_O$ because they bind more favorably to O; green, agonists also stabilize the gating transition state (‡) to increase the opening rate constant because the LA→HA induced fit ('hold', **Figure 2**) occurs at the start of the gating conformational change. Calibration: ACh, adult-type AChRs, –100 mV, 23 °C. (**B**) Corresponding reaction scheme (2 equivalent and independent neurotransmitter sites). Horizontal, ligand-protein complex formation; $K_{dC}$ and $K_{dO}$, LA and HA equilibrium dissociation constants to C and O. Vertical, global conformational change; $L_n$, gating equilibrium constant with n bound agonists. Red dashed line, pathway that determines the CRC when $L_0$ is negligible (**Equation 4a**, **Equation 4b**, **Equation 4c**, **Equation 4d**, Methods).

In adult-type AChRs and without any bound agonists, the probability of the O conformation ($P_O$) is small (~$10^{-6}$) so baseline current is negligible (**Jackson, 1986**; **Nayak et al., 2012**; **Purohit and Auerbach, 2009**). However, when both agonist sites are occupied by the neurotransmitter ACh, the increase in favorable binding free energy realized in the spontaneous isomerization from C to O increases $P_O$ substantially (to ~0.96), to generate a cellular current (**Figure 1**).

Our approach to investigating the AChR allosteric transition is to measure free energy changes associated with each component event. Energy and structure are related, and a longer term goal is to associate these energy changes with those in structure. Briefly, we use electrophysiology to compile a CRC and from the minimum, maximum and midpoint calculate the ligand's LA and HA equilibrium dissociation constants for binding to C and to O ($K_{dC}$ and $K_{dO}$). The logs of these are proportional to the LA and HA binding free energies, $\Delta G_{LA}$ and $\Delta G_{HA}$. As shown below, agonist efficiency depends only on these two values.

Previously, it was observed that in AChRs the ratio $\Delta G_{LA}/\Delta G_{HA}$ is approximately the same for a group of agonists despite wide variation in individual affinities and efficacies (**Jadey and Auerbach, 2012**). Within this group, whose members included full agonists (like ACh), partial agonists (like nicotine) and extremely weak agonists (like choline), the binding free energy ratio was 0.51±0.01 (mean ± sd). That is, for all members in the group binding to O is ~two-fold stronger than to C, regardless of agonist affinity or efficacy. Additional experiments revealed two more groups having a shared binding energy ratio, either 0.58 (for example, epibatidine) or 0.46 (for example, tetramethylammonium) (**Indurthi**

**Table 1.** Agonist efficiencies.

| Agonist | EC$_{50}$ (µM) (sem) | P$_o^{max}$ (sem) | K$_{dC}$ (µM) | K$_{dO}$ (nM) | c | η | n |
|---|---|---|---|---|---|---|---|
| BzTMA[a] | 1070 (200) | 0.60 (0.05) | 930 (112.0) | 650 (90) | 1424 (18.0) | 0.51 (0.01) | 3 |
| Dec[b] | 190 (20) | 0.79 (0.03) | 90 (7.0) | 140 (10) | 643 (21) | 0.41 (0.01) | 4 |
| SCh[a] | 20 (3) | 0.84 (0.02) | 50 (4.0) | 20 (1) | 3177 (118) | 0.45 (0.01) | 3 |
| BzTEA[b+c] | 2 (0.1) | 0.85 (0.02) | 0.80 (0.3) | 3 (2) | 316 (14) | 0.29 (0.00) | 3 |
| TriMAa[b] | 16000 (1200) | 0.78 (0.03) | 7720 (316.0) | 12580 (670) | 615 (18) | 0.57 (0.01) | 4 |

Mean EC$_{50}$ and P$_o^{max}$ were measured from each CRC (intra-cluster interval duration histograms in **Figure 3** and **Figure 3—figure supplement 1**), with standard error of mean (sem). K$_{dC}$ and K$_{dO}$ were calculated (**Equation 4a**) after correcting the background mutations that only changed L$_0$. c, coupling constant (K$_{dC}$/K$_{dO}$); η, efficiency (**Equation 2**); error estimates (calculated by error propagation) for calculated K$_{dC}$, K$_{dO}$, c, and η values given by (sem); n, number of CRCs. Membrane potential, +70 mV (to minimize channel block by the agonist). Agonist structures are in **Figure 5**; superscripts indicate mutation backgrounds: [a]εS450W, [b]εL269F, [c]εE181W that increase L$_o$ (increase responses to weak agonists).

The online version of this article includes the following source data for table 1:

**Source data 1.** Agonist efficiency.

and Auerbach, 2021; Nayak et al., 2019). In these groups, agonist binding free energy increases by ~1.7 and ~2.2-fold in the channel-opening conformational change.

The observation of a common ΔG$_{LA}$/ΔG$_{HA}$ ratio for even one group of ligands is remarkable because it implies that the only 2 events in receptor activation that involve the agonist directly - LA binding to C and the switch to HA that happens within gating - are not independent. Rather, the constant ratio indicates that the energy (structure) changes that underpin LA binding to C and HA binding to O are related (**Auerbach, 2016**; **Jadey and Auerbach, 2012**; **Purohit et al., 2014**).

Below, we define η empirically as the receptor's output/input energy ratio, and propose that in AChRs it is the fraction of agonist binding energy applied to the mechanical work of a local rearrangement of the neurotransmitter site (an induced fit) that initiates the gating conformational cascade (**Nayak et al., 2019**). In principle, η can be calculated from any CRC to estimate the degree of coupling between the maxima of receptor output (efficacy) and agonist input (affinity).

We measured η for various agonists of AChRs, both wild-type (wt) and following mutation of a binding site residue. Here we report 16 new values (shown in **Tables 1 and 2**) that, combined with 60 previous measurement describe a spectrum of 5 η classes. The presence of multiple η classes obscures the underlying correlations between affinity and efficacy and, further, suggests that there are multiple C versus O binding site structures. Importantly, the existence of efficiency classes highlights that binding and gating are energy-linked stages of a unified allosteric transition.

# Results
## Background and definitions

The standard conception of receptor activation incorporates two seemingly disparate events – bind, the formation of a ligand-protein complex, and gate, the global isomerization of the protein (**Figure 1**). However, as described below (Figure 2), in AChRs these are composite reactions and are connected by a pair of local, induced-fit rearrangements of the agonist site (**Jadey and Auerbach, 2012**).

For clarity, we define the universal agonist attributes affinity and efficacy. Affinity is the strength at which the ligand binds to its target site. In receptors ligands have two affinities, weak binding to C and strong binding to O (Figure 1). The corresponding binding free energies, ΔG$_{LA}$ and ΔG$_{HA}$, are calculated (in kcal/mol) as +RT times the natural logarithms of the apparent K$_{dC}$ and K$_{dO}$, where R is the gas constant and T is the absolute temperature (RT = 0.59 at 23 °C). For ACh at adult-type binding sites K$_{dC}$ = 174 µM (ΔG$_{LA}$ = −5.1 kcal/mol) and K$_{dO}$ = 29 nM (ΔG$_{HA}$ = −10.2 kcal/mol) (**Jadey and Auerbach, 2012**).

**Table 2.** Mutation efficiencies.

| mutation | agonist | EC$_{50}$ (μM) | P$_O^{max}$ | K$_{dC}$ (μM) | K$_{dO}$ (nM) | c | η$^{mut}$ | n | η$^{wt}$ |
|---|---|---|---|---|---|---|---|---|---|
| D200A | ACh[a] | 110 (10) | 0.80 (0.02) | 52 (3) | 150 (5) | 338 (12) | 0.37 (0.00) | 3 | 0.50 |
| | CCh[a] | 320 (70) | 0.45 (0.02) | 138 (18) | 900 (94) | 153 (3) | 0.36 (0.00) | 4 | 0.52 |
| | TMA[a] | 13350 (2970) | 0.37 (0.03) | 5683 (740) | 4382 (890) | 130 (5) | 0.48 (0.02) | 5 | 0.54 |
| | Ebt[a+b] | 110 (20) | 0.50 (0.03) | 47 (5) | 680 (47) | 69 (2) | 0.30 (0.00) | 3 | 0.41 |
| | Ebx[a+b] | 060 (20) | 0.49 (0.03) | 26 (5) | 390 (58) | 67 (3) | 0.29 (0.01) | 3 | 0.46 |
| K145A | ACh[a] | 110 (10) | 0.95 (0.06) | 83 (5) | 50 (3) | 1672 (82) | 0.44 (0.00) | 3 | - |
| | CCh[a] | 380 (30) | 0.53 (0.01) | 165 (8) | 400 (15) | 411 (5) | 0.41 (0.00) | 4 | - |
| | TMA[a] | 2200 (590) | 0.19 (0.01) | 922 (125) | 5010 (712) | 184 (3) | 0.43 (0.01) | 3 | - |
| | Ebt[a] | 40 (4) | 0.81 (0.04) | 20 (2) | 20 (2) | 792 (52) | 0.38 (0.00) | 3 | - |
| | Ebx[a] | 40 (3) | 0.83 (0.01) | 20 (1) | 20 (1) | 847 (17) | 0.38 (0.00) | 3 | - |
| G153S | Ebt[a] | 2 (0.5) | 0.65 (0.01) | 2 (0.3) | 7 (1) | 313 (4) | 0.31 (0.01) | 4 | - |

Measured EC$_{50}$ & P$_O^{max}$ and calculated K$_d$, c and h, mean (sem). For mutation location see **Figure 6**, inset. K$_{dC}$ and K$_{dO}$ were calculated from CRC parameters (**Figure 6**) by using Eq. 4. c, coupling constant (K$_{dC}$/K$_{dO}$); n, number of CRCs. L$_0$ was corrected for background mutations (Methods): [a]εS450W, [b]εL269F, [c]εE181W. See **Figure 6—figure supplement 1** and **Figure 6—figure supplement 2** for intra-cluster interval duration histograms.

The online version of this article includes the following source data for table 2:

**Source data 1.** Mutation efficiency.

Efficacy can be defined in several ways. Oftern, it is simply the high-concentration asymptote (maximum response) of an unnormalized CRC that in our experiments is P$_O^{max}$. Considering just bind and gate (**Figure 1B**), this limit depends only on the fully-liganded gating equilibrium constant L$_2$ so this constant, too, defines agonist efficacy (**Equation 4a**). Another definition derives from considering the full cycle of receptor activation. In adult-type AChRs the 2 neurotransmitter binding sites are approximately equivalent and independent and there is no significant input of external energy (**Nayak and Auerbach, 2017**), so

$$\frac{L_2}{L_0} = \left(\frac{K_{dC}}{K_{dO}}\right)^2. \tag{1}$$

The subscripts of the gating equilibrium constants (L) refer to the number of bound agonists, and the equilibrium dissociation constant (K$_d$) ratio is called the coupling constant (c). L$_0$ is agonist-independent so differences in L$_2$ (efficacy) among agonists depend only on differences in c. Below, we use the logarithm of c ($\lambda$) as the index of relative agonist efficacy,

$$\lambda = \Delta G_{HA} - \Delta G_{LA}.$$

The relative efficacy of an agonist depends only on the difference between binding free energies, O minus C (blue lines in **Figure 1A**).

Unlike voltage and mechanical stimuli, a small, thermalized ligand can deliver only a small force to a large receptor (**Howard, 2001**). The tiny momentum imparted to the protein by the ligand is obscured by those from collisions with water molecules. In the absence of external energy, agonists promote conformational change only by providing more favorable (stabilizing) binding energy to

active compared to resting states that interconvert spontaneously. This mechanism likely pertains to all large receptors activated by small agonists.

Efficiency ($\eta$) is defined empirically as the maximum output/input energy ratio (the efficacy/high-affinity energy ratio),

$$\eta = \frac{(\Delta G_{HA} - \Delta G_{LA})}{\Delta G_{HA}}$$
$$= 1 - \frac{\Delta G_{LA}}{\Delta G_{HA}} \tag{2}$$

Accordingly, the agonist-dependent free energy changes in gating ($\Delta G_{HA}$-$\Delta G_{LA}$) and in binding ($\Delta G_{HA}$) are interchangeable, with $\eta$ as the conversion factor. Again, efficacy ($\lambda$) is a free energy *difference* and the maximum amount the agonist can deliver to the receptor's gating machinery, and efficiency ($\eta$) is a free energy *ratio* and this amount normalized by the ligand's maximum binding energy. With

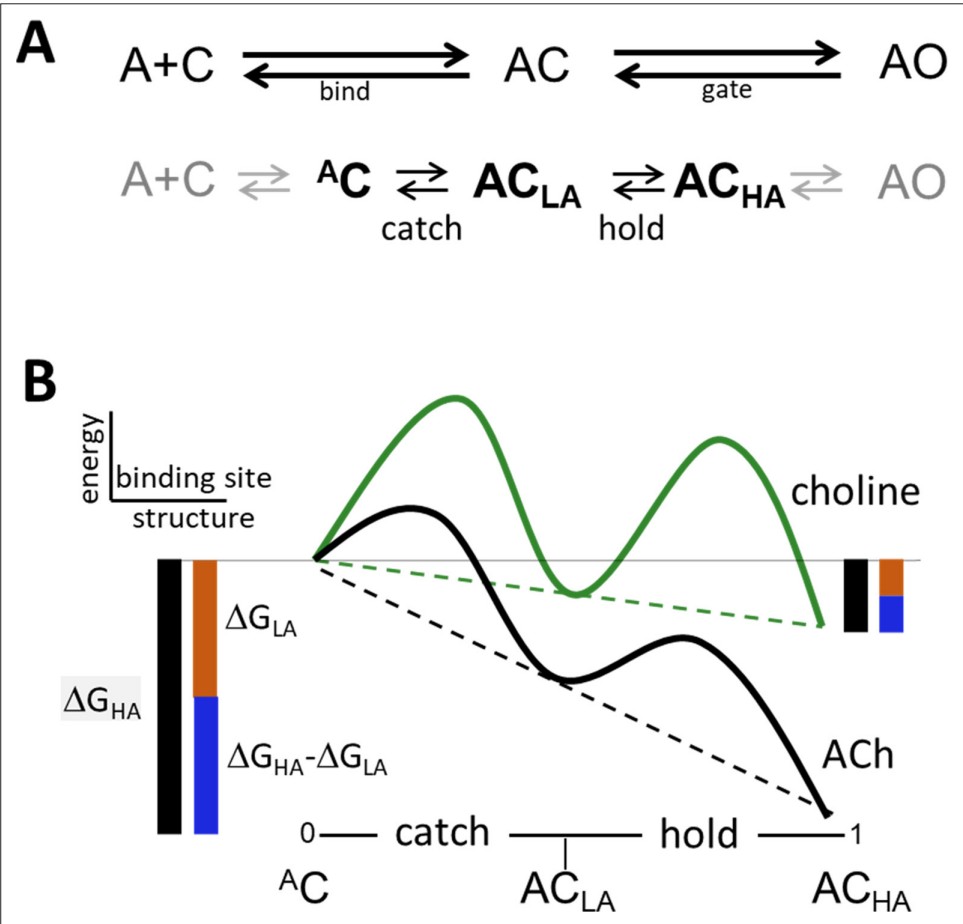

**Figure 2.** Catch and hold. (**A**) Bind and gate. Top, as one-step reactions. Bottom, as composite reactions. The main undetected intermediate states are $^A$C, the encounter complex (inside bind) and AC$_{HA}$, a high affinity and closed channel state (inside gate). Black, the 2 agonist-dependent induced fit rearrangements are called catch ($^A$C$\rightleftarrows$AC$_{LA}$) and hold (AC$_{LA}$$\rightleftarrows$AC$_{HA}$). Gray, diffusion (A+C$\rightleftarrows$$^A$C) and receptor conformation changes in other domains (AC$_{HA}$$\rightleftarrows$AO) are approximately agonist-independent (see *Figure 2—figure supplement 1*). (**B**) Catch-and-hold free energy landscape. An $\eta$ class indicates that catch and hold are linked in a linear free energy relationship (LFER; dashed lines) (*Howard, 2001*). Green, weak agonist; black, strong agonist. Ligand binding energy 'tilts' the entire landscape to the total extent $\Delta G_{HA}$ (black side bars). For relative agonist actions, the energy change in catch (brown) determines K$_{dC}$ and the energy change in hold (blue) determines the coupling constant. $\eta$ is the fraction of the total applied to hold (blue/black) and $\eta$ is the fraction aplied to catch (brown/black). Agonist affinity and efficacy differ substantially, but $\eta$ is constant and depends only on the left-right position of AC$_{LA}$ in the reaction.

The online version of this article includes the following figure supplement(s) for figure 2:

**Figure supplement 1.** Inside bind and gate.

regard to equilibrium dissociation constants, $\lambda$ relates to $\log(K_{dC}/K_{dO})$ and $\eta$ relates to $\log(K_{dC})/\log(K_{dO})$ (*Figure 7—figure supplement 2*).

Insofar as $K_{dC}$ and $K_{dO}$ are universal agonist attributes, so too is $\eta$. We propse that an $\eta$ value can be calculated from the ratio of logarithms of the 2 dissociation constants for every agonist of every receptor. However, this does not imply that $\eta$ has a physical meaning, or that all agonists of a given receptor have the same (or a unique) $\eta$ value. *Equation 2* only shows how to calculate $\eta$ from the 2 equilibrium dissociation constants. We estimated these from CRCs, but other experimental approaches would suffice.

In AChRs $\eta$ does have a physiochemical meaning because $K_{dC}$ and $K_{dO}$ derive mainly from a pair of induced fit rearrangements at the ligand site. Although bind and gate are usually denoted as single-step events (*Figure 1B*), in AChRs both are composite reactions that harbor intermediate states that are too short-lived to be detected individually and directly (*Figure 2*, *Figure 2—figure supplement 1*). Induced fits are common, and undetected intermediate states have been invoked with great success previosuly to explain experimental results (for example ES in enzymology, and AC in pharmacology). Here, we invoke intermediate states to deconstruct $\eta$ (*Figure 2*).

In bind, the agonist diffuses to the target and forms an encounter complex (*Held et al., 2011*; *Homans, 2007*; *Schiebel et al., 2018*), after which a local rearrangement (an induced fit) called 'catch' establishes the LA complex ($A+C \rightleftarrows {}^AC \rightleftarrows AC_{LA}$). Gate starts with the second stage of the induced fit called 'hold' that increases agonist affinity and, after several additional conformational changes in other protein domains, terminates with rearrangements in the pore that allow water and ions to pass ($AC_{LA} \rightleftarrows AC_{HA} \rightleftarrows \ldots \rightleftarrows AO_{HA}$).

According to *Equation 2*, efficiency depends only on the LA/HA binding energy ratio. Regarding $\Delta G_{LA}$ (proportional to $\log K_{dC}$), the agonists we examined (Figure 5) all have approximately the same diffusion constant so we surmise that differences are caused by differences in catch. Regarding $\Delta G_{HA}$ (proportional to $\log K_{dO}$), the effect of perturbations away from the neurotransmitter site are agonist independent so we surmise that differences are caused by differences in hold. Therefore, with regard to $\eta$ and the CRC (red pathway, *Figure 1B*), we attribute differences between agonists exclusively to differences in the free energy changes associated with induced fit that is the pair of catch-hold rearrangements, ${}^AC \rightleftarrows AC_{LA} \rightleftarrows AC_{HA}$ (*Figure 2B*).

The energy change in catch is $\Delta G_{LA}$ and in hold is $\Delta G_{HA}-\Delta G_{LA}$. Their sum, $\Delta G_{HA}$, is the total free energy change delivered by the ligand. From *Equation 2*, $\eta$ is the fraction of this total associated with hold, and 1-$\eta$ is the fraction associated with catch. Multiplying $\eta$ by 100% gives the percent of agonist binding energy used for the hold rearrangement of the binding site that jumpstarts the full conformational cascade that connects the neurotransmitter sites with the gate.

In a pure binding reaction, for example $A+C \rightleftarrows {}^AC$, $K_d$ is the concentration where the energy gained from formation of the complex is equal to the entropy lost from removing a ligand from solution that is indexed to a reference concentration (*Phillips, 2020*). However, if $K_{dC}$ and $K_{dO}$ are dominated by free energy changes associated with the induced fit, the entropy components become negligible (*Figure 7—figure supplement 1*). As far as comparative agonist action in AChRs is concerned, it appears that only the energy changes in catch and hold are germane.

$EC_{50}$ and $P_o^{max}$ were measured from each CRC (intra-cluster interval duration histograms in *Figure 3* and *Figure 3—figure supplement 1*). $K_{dC}$ and $K_{dO}$ were calculated (*Equation 4a*, *Equation 4b*, *Equation 4c*, *Equation 4d*) after correction for background mutations that only changed $L_0$. c, coupling constant ($K_{dC}/K_{dO}$); $\eta$, efficiency (*Equation 2*); n, number of CRCs. Membrane potential,+70 mV (to minimize channel block by the agonist). Agonist structures are in Figure 5; superscripts indicate mutation backgrounds: [a]εS450W, [b]εL269F, [c]εE181W that increase $L_o$ (to increase responses to weak agonists).

## Agonists

We measured $\eta$ for 5 agonists using adult-type AChRs with wt neurotransmitter binding sites (*Table 1*). For each, single-channel currents were recorded at different agonist concentrations and $P_O$ values calculated from shut and open interval durations were compiled into a CRC. $L_0$ was known a priori (*Nayak et al., 2012*) so $K_{dC}$ and $K_{dO}$ could be calculated from $EC_{50}$ and $P_O^{max}$ by using *Equation 4a*, *Equation 4b*, *Equation 4c*, *Equation 4d* (Materials and methods). As shown elsewhere, CRCs compiled from whole-cell currents serve equally well for $\eta$ estimation (*Indurthi and Auerbach, 2021*).

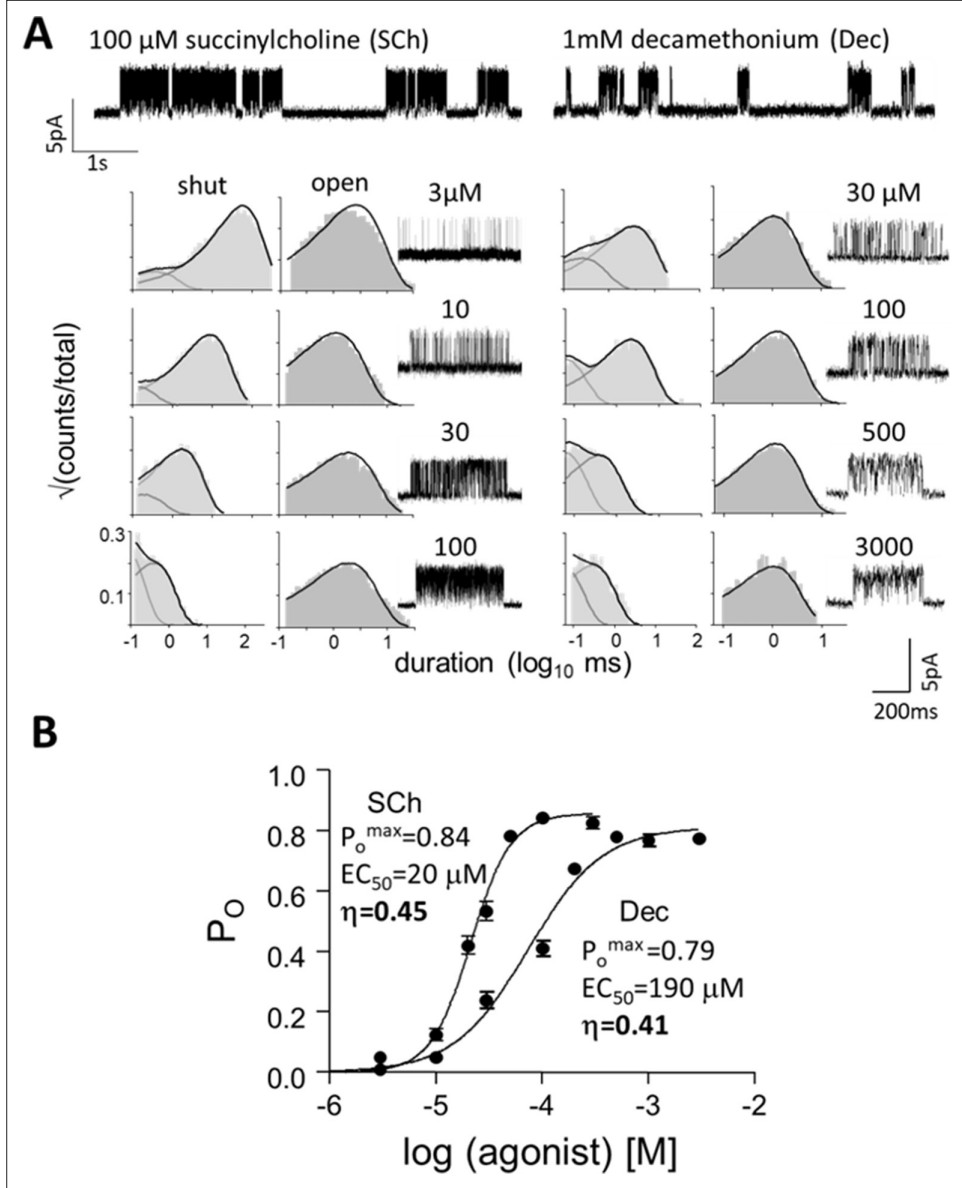

**Figure 3.** Measuring $\eta$. (**A**) Top, single-channel current traces, low time-resolution ($V_m$ = +70 mV; O is up). Clusters of openings are bind-and-gate (**Figure 1B**); silent periods between clusters are desensitized. Bottom, example clusters and intra-cluster interval duration histograms. $P_O$ was calculated at each [agonist] from shut- and open-interval time constants. (**B**) CRCs. $P_O$ values were fitted to estimate $P_O^{max}$ and $EC_{50}$ from which $K_{dC}$ and $K_{dO}$ were calculated (**Equation 4a**, **Equation 4b**, **Equation 4c**, **Equation 4d**, Materials and methods) (**Table 1**). The logs of these constants are proportional to $\Delta G_{LA}$ and $\Delta G_{HA}$, the ratio of which gives $\eta$ (**Equation 2**). The profile for SCh is relatively left-shifted because this agonist is ~10% more efficient than Dec. Symbols are mean ± sem (**Table 1**; see **Figure 3—figure supplement 1** for other agonists). The background mutation εS450W compensates for the effects of depolarization on the gating rate constants (see Materials and methods).

The online version of this article includes the following figure supplement(s) for figure 3:

**Figure supplement 1.** CRCs.

*Figure 3* shows CRCs for succinylcholine (SCh) and decamethonium (Dec). Although $P_O^{max}$ is similar for both agonists, $EC_{50}$ (potency) of SCh is substantially lower than that of Dec. For each agonist, $\Delta G_{LA}$ and $\Delta G_{HA}$ were calculated to yield an $\eta$ value (*Equation 2*) with the result were $\eta_{SCh}$=0.45 and $\eta_{Dec}$=0.41 (*Table 1*). SCh is 10% more efficient than Dec, which is the root cause of the abovementioned mismatch between $P_O^{max}$ and potency. *Figure 3* shows that given $L_0$, agonist efficiency can be

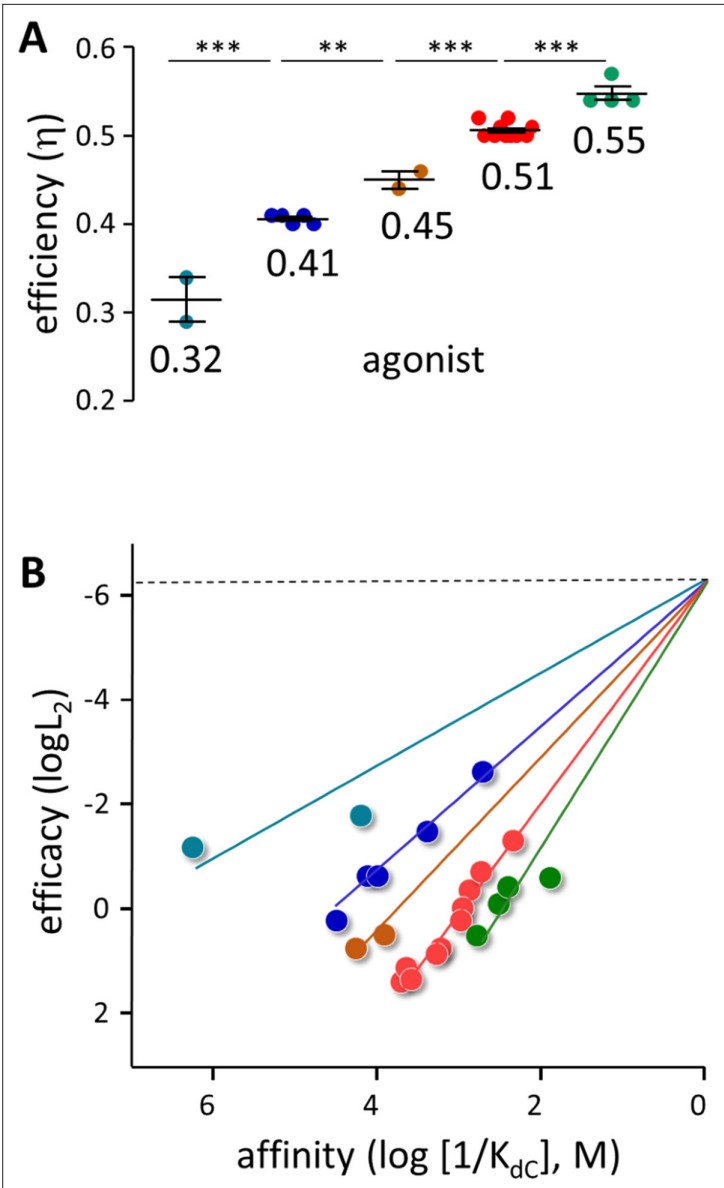

**Figure 4.** Agonist h values. (**A**) Each symbol represents the average $\eta$ value of one agonist, calculated from CRCs (*Table 1*, *Table 1—source data 1*). The x-means algorithm was used to separate the agonists into 5 $\eta$ classes (mean, vertical bars are ± sd). (**B**) Efficiency plots. Each color group was fitted by a straight line (*Equation 3*) with $L_0 = 5.2 \times 10^{-7}$ (sd of each point smaller than the symbol). $\eta$ values calculated from the slopes are the same as in panel A. x- and y-axes are proportional to the agonist's free energy changes in catch ($\Delta G_{LA}$) and hold ($\Delta G_{HA} - \Delta G_{LA}$) induced fits (*Figure 2B*).

calculated from a single CRC. Because $\eta$ is the ratio of two logarithms, error arising from errors in $P_O^{max}$ and $EC_{50}$ are small (*Indurthi and Auerbach, 2021*). CRCs for the other agonists are in *Figure 3—figure supplement 1*.

$\eta$ values were calculated from previous measurements of $K_{dC}$ for 18 other agonists (*Figure 4A*). An x-means cluster analysis of all 23 agonist $\eta$ values indicates that there are 5 groups (mean ± sd): 0.32±0.035, 0.41±0.005, 0.45±0.014, 0.51±.008 and 0.55±0.015.

*Figure 5* shows the agonists grouped by $\eta$ class. The neurotransmitter ACh, its breakdown product choline (Cho), and the partial agonists carbamylcholine (CCh) and nicotine all belong to the $\eta$~0.51 class. The second-most common class, $\eta$~0.41, includes ligands that have a bridge nitrogen (for instance, Ebt) as well as others that do not (for instance, Dec and TMP). Small ligands (for instance,

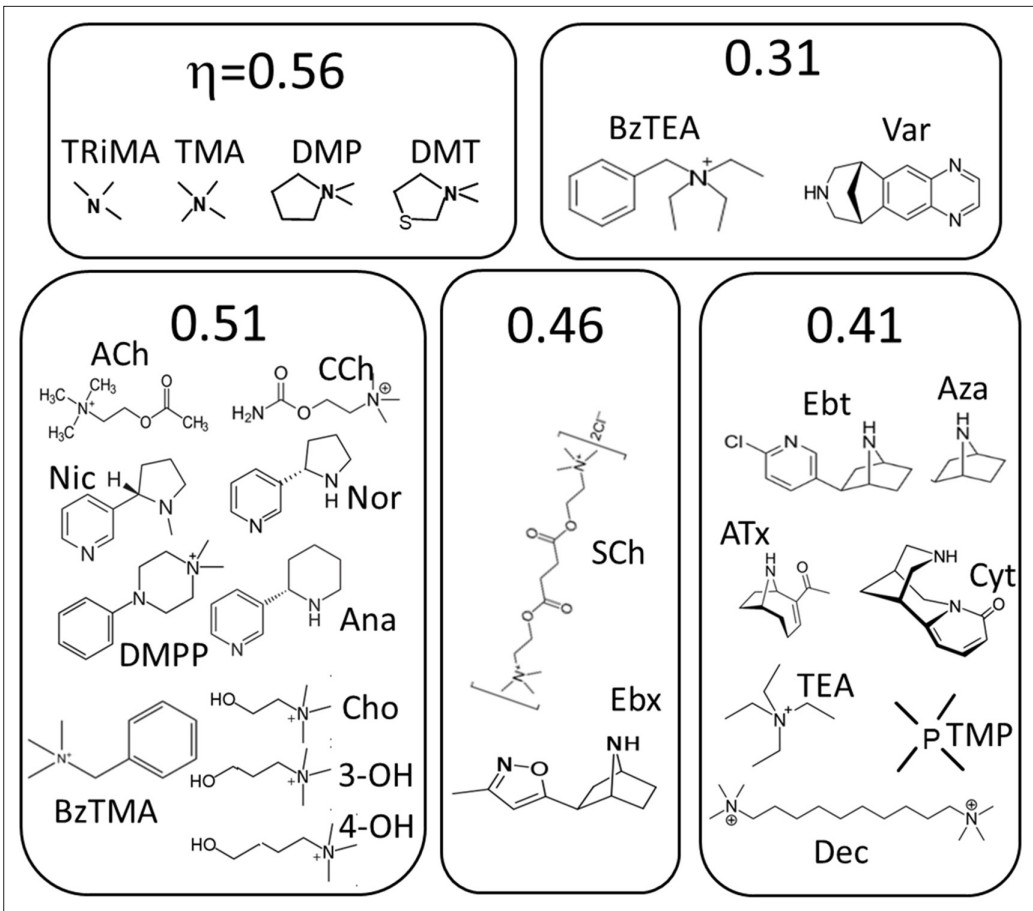

**Figure 5.** Agonists grouped by $\eta$ class (**Figure 4**). See Materials and methods for abbreviations.

TriMA, MW 60) appear to be the most efficient and large rigid ligands (for instance, varenicline, MW 211) appear to be the least efficient.

It is valuable to combine Equations 1 and 2 (**Nayak et al., 2019**),

$$\log L_2 = \log L_0 + m \log(\frac{1}{K_{dC}})$$
$$\eta = \frac{m}{(m+2)}.$$

(3)

*Equation 3* describes an 'efficiency' plot, log affinity ($1/K_{dC}$) versus log relative efficacy ($L_2$). An average $\eta$ for a group of agonists is estimated from the slope of the straight line fit (m). *Equation 4a*, *Equation 4b*, *Equation 4c*, *Equation 4d* converts readily measured CRC parameters ($P_O^{max}$ and $EC_{50}$) into equilibrium constants ($K_{dC}$ and $L_2$), and *Equation 3* converts these into fundamental constants that pertain to the agonist ($\eta$) and the receptor ($L_0$). The value of the efficiency plot is that it increases the accuracy of the $\eta$ estimates because if $L_0$ is known a priori, the y-intercept can be added as a fixed point to all lines. For receptors in which $L_0$ has not been measured, the efficiency plot offers a convenient way to do so, as was done previously for glutamate, GABA, glycine, and muscarinic recepotors (**Nayak et al., 2019**).

$\eta$ values estimated from the slopes for the 23 agonists (**Figure 4B**) are the same as from the x-mean cluster analysis (**Figure 4A**) (mean ± sd): 0.31±0.018, 0.41±0.002, 0.46±0.004, 0.51±0.002 and 0.56±0.008. Affinity and efficacy are correlated significantly within the 3 classes having >2 members (19 agonists; Pearson's correlation test p-values for $\eta$=0.41 and 0.51,<0.0001, and for $\eta$=0.56, 0.019). Also, the slopes for these 3 classes are significantly different (ANCOVA, *F*=129, p<0.001). The association of the other agonists with a discrete $\eta$ class is less certain but supported by mutation studies (see below).

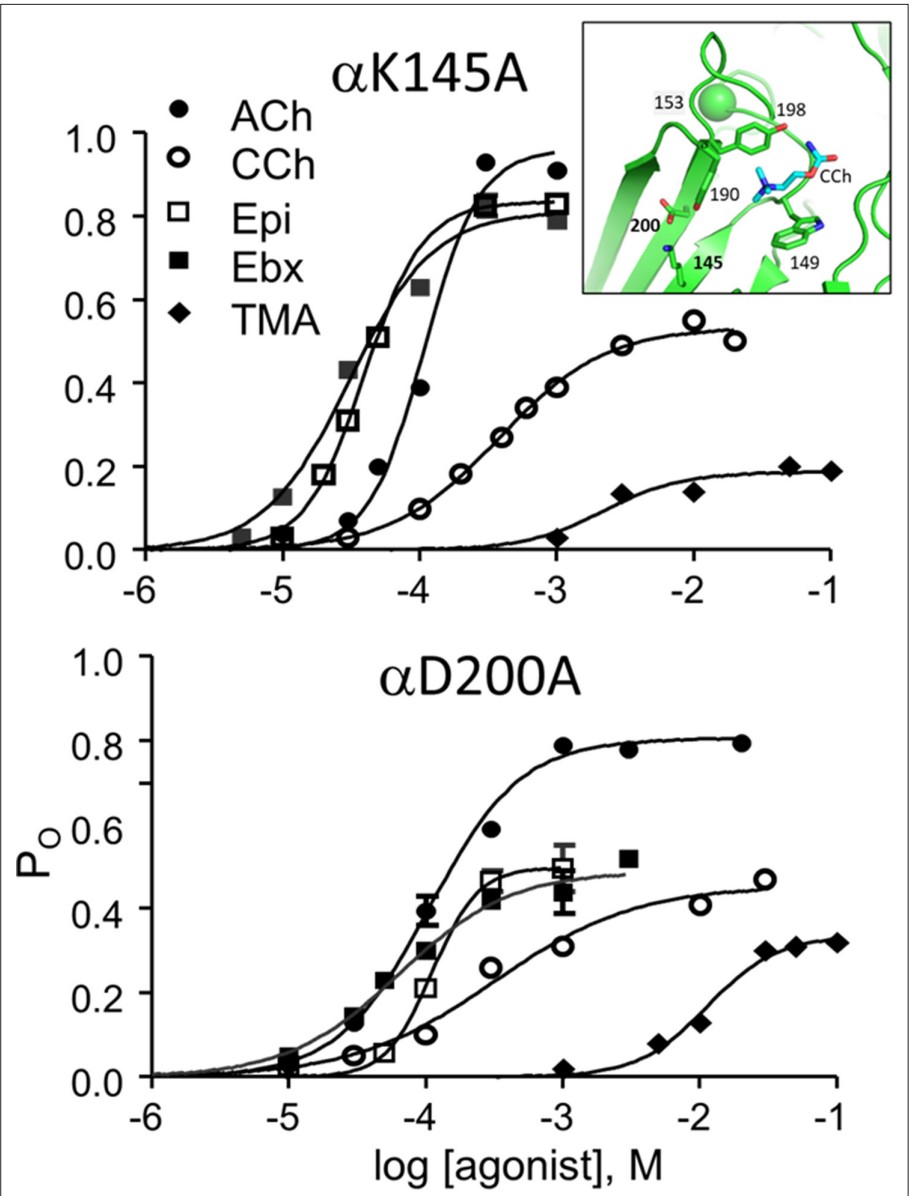

**Figure 6.** CRCs for αK145A and αD200A. Agonist efficiencies calculated from the fitted CRC parameters are in *Table 2*. Intra-cluster interval histograms are in *Figure 6—figure supplement 1* and *Figure 6—figure supplement 2*. Inset, α–δ subunit interface of an AChR neurotransmitter binding site occupied by CCh (cyan) (7QL6.pdb; *Zarkadas et al., 2022*).

The online version of this article includes the following figure supplement(s) for figure 6:

**Figure supplement 1.** αK145A.

**Figure supplement 2.** αD200A and αG153S.

*Figure 4B* shows that in AChRs, agonists having the same affinity can have different efficacies, and *vice versa*. Absent classification of an agonist according to $\eta$ (imagine all symbols the same color) there is no global correlation between these two agonist properties. However, a linear correlation between log affinity and log efficacy is clear within each class. In AChRs, the presence of multiple $\eta$ classes precludes a global correlation between efficacy and affinity.

## Mutations

$\eta$ values were measured previously in adult-type AChRs having one of 42 binding site mutations (*Table 2—source data 1*). To these we add 11 more (*Table 2*), for 5 agonists and 3 binding-site

mutations (αD200A, αK145A, αG153S; *Figure 6*, *Figure 6—figure supplement 1*, *Figure 6—figure supplement 2*).

αD200 and αK145, along with αY190, have been suggested to work together to initiate the channel-opening conformational change (*Mukhtasimova et al., 2005*). The mutation αY190A reduces $\eta_{ACh}$ from 0.50 to 0.35, but αY190F is without effect (*Bruhova and Auerbach, 2017*). However, αY190F does cause substantial losses in LA and HA binding energies for ACh, to an extent that depends on the αK145 side chain (*Bruhova and Auerbach, 2017*). These results support the suggestion that these side chains work together, but exactly how and to what effect remains unclear. The agonists we tested with αD200A or αK145A were from 4 different $\eta$ classes (wt class value): TMA (0.54), CCh (0.51), ACh (0.50), Ebx (0.46), and Ebt (0.41).

For mutation location see *Figure 6*, inset. $K_{dC}$ and $K_{dO}$ were calculated from CRC parameters (*Figure 6*) by using *Equation 4c*, coupling constant ($K_{dC}/K_{dO}$); n, number of CRCs. $L_0$ was corrected for background mutations (Materials and methods): [a]εS450W, [b]εL269F, [c]εE181W. See *Figure 6—figure supplement 1* and *Figure 6—figure supplement 2* for intra-cluster interval duration histograms.

The results are in *Table 2*. The substitution αD200A reduces the efficiency of 4 of the 5 agonists to 0.33±0.04 (mean ± sd), or to about the same level as with αY190A with ACh. The exceptional ligand was TMA for which $\eta$ remained relatively high at 0.48. In contrast, in αK145A $\eta$ for all 5 agonists was 0.41±0.03. Interestingly, this mutation reduces $\eta$ for ACh, CCh and TMA but not significantly for Ebx and Ebt. Note that the mutation αW149A increases $\eta$ for ACh to 0.60 (*Purohit et al., 2014*).

*Table 2—source data 1* shows $K_{dC}$ and $K_{dO}$ values measured previously for 4 different agonists in AChRs having a substitution at αG153 (*Jadey et al., 2013*), including a Ser that causes a congenital myasthenic syndrome (*Engel et al., 1982*). After converting the published equilibrium constants to $\Delta G_{LA}$ and $\Delta G_{HA}$, the results indicate that on average the mutations reduce $\eta$ for Cho, DMP, TMA and nicotine to 0.41±0.03 (mean ± sd), or by ~25%. To these we add the new result that αG153S reduces $\eta$ of Ebt from 0.42 to 0.31, or also by ~25%.

*Figure 7A* shows x-means cluster analysis of efficiencies for 53 AChR binding site mutations. Mutant $\eta$ values (mean ± sd) segregate into the same 5 $\eta$ classes that were apparent with agonists: 0.33±0.03, 0.40±0.02, 0.44±0.01, 0.49±0.01 and 0.56±0.03. The $\eta$ classes that were under-represented and poorly defined with agonists (*Figure 4B*) are more common and clearer with mutations.

With mutations, an $\eta$-plot is not useful because substitutions can change $L_0$ (see *Figure 7—figure supplement 1*) and, hence, the y-intercept of each line. Instead, a plot of $\Delta G_{LA}$ versus $\Delta G_{HA}$ shows the binding energy correlations directly. This plot for mutations (*Figure 7B*) shows 5 slopes (Pearson's correlation test p-value <0.0006 for all classes). The distribution of $\eta$ values (1-slope) is (mean ± sd): 0.32±0.007, 0.40±0.004, 0.44±0.003, 0.49±0.002 and 0.56±0.007, with all slopes being significantly different (ANCOVA, F-value, 63.35; p-value,<0.0001).

Combining the results for agonists and mutations, the overall distribution of $\eta$ (relative prevalence) is: 0.56 (17%), 0.51 (31%), 0.45 (13%), 0.41 (26%), and 0.31 (12%). As was the case with agonists and mutations separately, the 0.51 (example, ACh) and 0.41 (example, Ebt) classes predominate. *Figure 7C* shows the distribution of agonist plus mutation $\eta$ values as a spectrum in which line thickness represents relative prevalence.

## Discussion

### Efficiency

Efficiency is the missing link that connects binding to gating (*Equation 2*). Insofar as $K_{dC}$ and $K_{dO}$ apply generally (*Figure 1*), $\eta$ is a universal agonist attribute that depends only on these two constants. In AChRs, $K_{dC}$ and $K_{dO}$ are set mainly by energy changes in a pair of local rearrangements of the binding site (the catch-and-hold induced fit), with $\eta$ being the fraction of the total used to initiate the allosteric transition of the receptor. As such, $\eta$ calibrates the fundamental connection that defines receptor action.

Converting ligand binding energy into energy for an otherwise unfavorable protein conformational change is an induced fit. We assume that without an agonist present, both catch and hold rearrangements (of an aromatic pocket) are energetically unfavorable and generate the high barrier to unliganded opening (Figure 1). In enzymes, a fraction of substrate binding energy is used to promote a local protein rearrangement that stabilizes the reaction transition state (*Richard, 2022*). In AChRs,

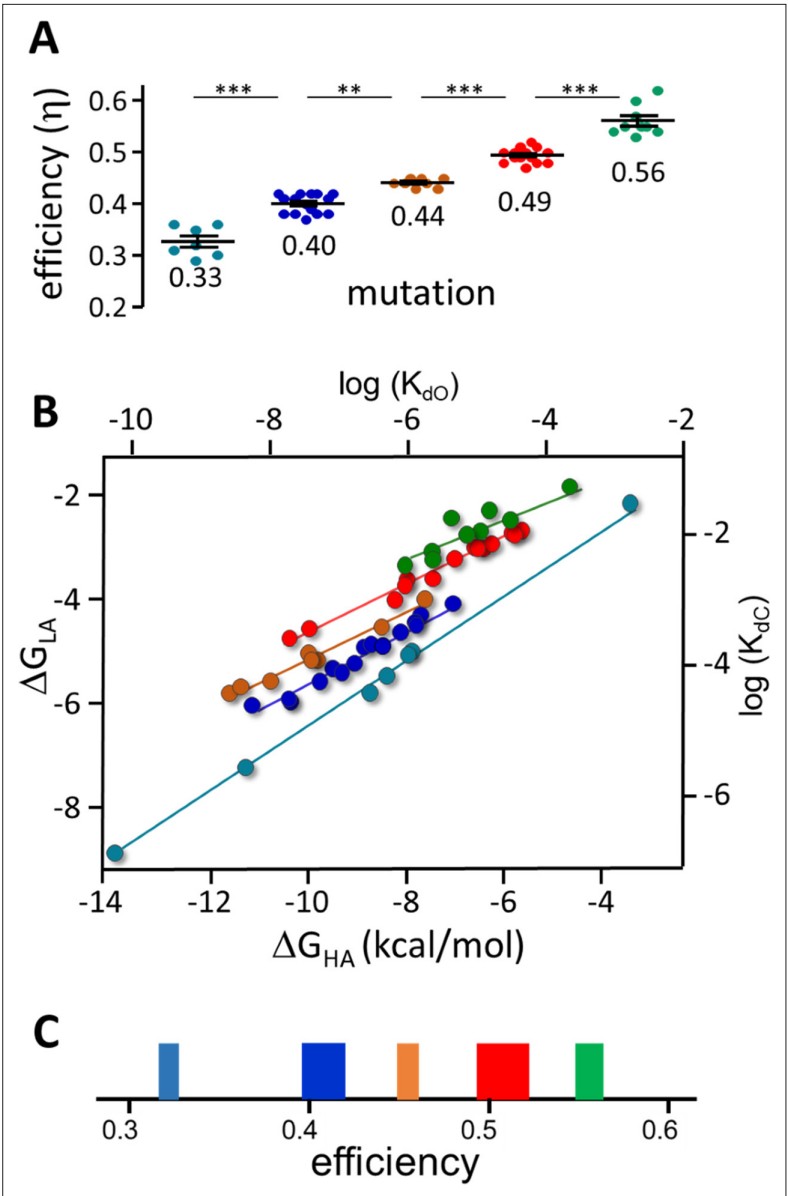

**Figure 7.** Mutation $\eta$ values. Each symbol is $\eta$ value calculated for an individual mutation (various agonists; *Table 2* and *Table 2—source data 1*). (**A**) There are 5 efficiency classes (mean ± sd). As with agonists (*Figure 4*), the ~0.5 and~0.4 classes predominate. (**B**) Weak (LA) versus strong (HA) binding energies for the mutants (sd of each point is smaller than the symbol). The 5 slopes reflect different correlations between catch and hold free energy changes (*Figure 2B*). The efficiency distribution for the mutations ($\eta$=1-slope; see text for mean ± sd) is the same as for agonists. C. Spectrum of efficiencies, adult AChR neurotransmitter binding sites. The width of each line is proportional to prevalence (agonists and mutations).

The online version of this article includes the following figure supplement(s) for figure 7:

**Figure supplement 1.** Estimating $L_0$ ($\alpha$D200).

**Figure supplement 2.** Evidence that in AChRs $K_{dC}$ reflects mainly the free energy change of the catch rearrangement (see *Figure 2—figure supplement 1*).

neurotransmitter binding energy is divided equally between two stages of an intrinsically unfavorable rearrangement that forms the LA and HA complexes and starts the gating isomerization. In brief, $\eta$ quantifies the split in ligand binding energy between the two steps in the catch-and-hold induced fit (*Figure 2B*).

## η classes

Agonists having radically different resting affinities and efficacies can have the same $\eta$. For example, ACh ($K_{dC}$ = 175 μM, $PO_m^{ax}$ = 0.96) and choline ($K_{dC}$ = 4 mM, $PO_m^{ax}$ = 0.05) both divide their binding energy equally between catch and hold. In AChRs, agonists segregate into discrete $\eta$ classes within which all members use approximately the same fraction of their post-diffusion binding energy for catch. So far we have identified 5 $\eta$ classes with catch percentages ranging from 47% to 71% (**Table 1**). Below, we discuss the possibility that larger ligands apply a larger percentage of their binding energy to catch.

$\eta$ classes are notable because its two component energy changes (Equation 3) are realized in bind and in gate, usually considered to be qualitatively different processes (**Figure 1B**). Nevertheless, the existence of an $\eta$ class indicates that in AChRs these two binding energy (structure) changes are correlated and joined in a linear free energy relationship (LFER) (**Figure 2B**). Any change in bind (catch) free energy compels one in gate (hold), reciprocally. Rather than being independent processes, bind and gate are completely entangled even if they are depicted as different dimensions of a reaction scheme (**Figure 1B**).

Although an empirical $\eta$ value can be calculated for every agonist of every receptor, the generality of $\eta$ classes is less certain. The components, $K_{dC}$ and $K_{dO}$ for many agonists, have not been measured extensively in other receptors, but some results suggest that a catch-and-hold induced fit, with stages linked in a LFER, is not uncommon.

Regarding hold, it is likely that an increase in affinity caused by a local binding site rearrangement inside the gating isomerization is typical ($AC_{LA} \rightleftarrows AC_{HA}$). In enzymes and receptors, agonists induce a 'clamshell' closure of the ligand site that is associated with increased binding strength (**Armstrong and Gouaux, 2000**; **Masiulis et al., 2019**; **Pless and Lynch, 2009**; **Traynelis et al., 2010**). Also, in many receptors agonists increase the opening rate constant as well as $P_O$ (**Clements et al., 1998**; **Clements and Westbrook, 1991**; **Maconochie et al., 1994**) indicating that the stabilization by the lgiadn of an otherwise unfavorable rearrangement (the affinity increase) occurs early in the isomerization (**Figure 1A**). Finally, in other receptors an $AC_{HA}$ intermediate state has been detected directly in single-channel currents (**Lape et al., 2008**; **Shi et al., 2023**) or identified in structures (**Sauguet et al., 2014**).

Regarding catch, this first step in the induced fit serves a purpose (**Figure 2—figure supplement 1**) and may also pertain to other receptors. As in AChRs, in several receptors $k_{on}$ to C is slower than diffusion and correlated with agonist potency (**Dravid et al., 2008**; **Grewer, 1999**; **Lewis et al., 2003**; **Mortensen et al., 2010**). Further, a catch-and-hold LFER is implied in other receptors (including a GPCR; **Sykes et al., 2009**) because efficiency plots are linear, with outliers that possibly indicate different $\eta$ classes (**Nayak et al., 2019**). In biological reactions, LFERs are empirical descriptors and more experiments are needed to determine the generality of $\eta$ classes. In addition to measurements of $K_{dC}$ and $K_{dO}$ in other receptors, and identifying the structures of intermediate liganded states $^A$C and $AC_{LA}$ would corroborate catch.

In AChRs, multiple $\eta$ classes together preclude the appearance of a global affinity-efficacy correlation. If $\eta$ classes turn out to be common, then the consensus view that there is no correlation between these agonist properties needs to be reexamined (**Colquhoun, 1998**; **Kenakin and Onaran, 2002**). The clear correlation between log affinity and log efficacy within each class vanishes when agonists from different classes are lumped together (**Figure 4B**).

We detected 5 $\eta$ classes in AChRs but additional experiments could reveal more. In particular, agonist classes with the highest and lowest $\eta$ values have the greatest scatter and could be amalgams. Nonetheless, the ability to corral 76 different agonist-receptor combinations into just 5 classes represents a significant reduction in complexity. If nothing else, $\eta$ is a useful way to classify agonists and, possibly, binding sites. Also, $\eta$ could prove to be useful CRC analysis because it allows $EC_{50}$ to be computed from $P_O^{max}$ and *vice versa* by using (**Equation 3**, **Equation 4a**, **Equation 4b**, **Equation 4c**, **Equation 4d**; **Indurthi and Auerbach, 2021**).

Mutations of binding site residues segregate into the same 5 efficiency classes as do agonists. This supports the idea that an agonist at a binding site is much like an ordinary side chain, with exceptions. Most importantly, agonists are not linked covalently and so are both hypermobile in the pocket and free to come and go to serve as a signal. Also, by definition a bound agonist is more stable in O versus C (**Figure 1A**) but this is the opposite of most AChR wt side chains (**Purohit et al., 2013**). $\eta$

measurements reinforce the standard view that ligands only perturb the intrinsic activity of allosteric proteins.

## η and structure

In AChRs, the pair of structural changes at the binding sites associated with energy changes in hold ($\Delta G_{HA}$-$\Delta G_{LA}$) and in catch ($\Delta G_{LA}$) are not known with certainty. Although there are structures of apo-C (*Zarkadas et al., 2022*) and desensitized (perhaps the same as $AO_{HA}$) (*Auerbach, 2020*; *Rahman et al., 2020*; *Zarkadas et al., 2022*), those of the three relevant liganded-closed intermediate states $^AC$, $AC_{LA}$ and $AC_{HA}$ are not available. Here, we discuss inferences about the structures of these three transient states that can be made from $\eta$ .

If every agonist had a unique $\eta$ then little could be said about the underlying conformational changes. However, the existence of an $\eta$ class and, therefore, an LFER between catch-and-hold energy changes suggests that the corresponding structural changes, too, are correlated, although perhaps not linearly. That is, catch and hold are two related stages of a single induced fit rearrangement of the binding site.

Regarding hold, at HA sites loop C covers ('caps') the aromatic pocket (*Nys et al., 2013*) that appears to be contracted. Also, simulations suggest the possibilities that in the hold step the ligand flips its orientation (*Tripathy et al., 2019*), and that water is extruded from the binding interface (*Young et al., 2007*). The components of the hold free energy change arise from these and other restructuring events, but the breakdown is not known. Regarding catch, little is known about this rearrangement directly because the structures of neither end state have been solved. (Note that apo-C and $^AC$ are different). However, because of the LFER, we hypothesize that all or some of the above putative rearrangements in hold (cap, contract, flip, de-wet) also happen in catch, but perhaps to lesser extents.

A surprising inference about the binding site structural change comes from measurements of agonist association rate constants ($k_{on}$) to C versus O (*Grosman and Auerbach, 2001*; *Nayak and Auerbach, 2017*). Whereas $k_{on}$ to C is slower than diffusion and correlated strongly with agonist potency, $k_{on}$ to O is approximately at the diffusion limit and weakly agonist dependent. Hence, slow association to C cannot be attributed to the ligand itself but rather suggests that the hold stage of the induced fit removes a barrier(s) that otherwise prevents free entry of the ligand into the aromatic binding pocket (*Figure 2—figure supplement 1*). Neither the site of the encounter complex nor the nature of this barrier are known. Because loop C capping and pocket contraction would be expected to decrease rather than increase $k_{on}$, we speculate that in AChRs other, still-unidentified structural changes in hold serve to remove the barrier. Removing loop C eliminates activation by agonists but not unliganded gating (*Purohit and Auerbach, 2009*). Apparently, clamshell closure is necessary for transducing ligand binding energy into the receptor isomerization, but not for the global isomerization itself. However, it is uncertain if other rearrangements in the binding site induced fit occur in unliganded gating after loop C removal (see below).

Energy change reflects a change in structure, so multiple $\eta$ classes ($\log K_{dC}/\log K_{dO}$ ratios) imply there are multiple pairs of C versus O (post-catch and post-hold) binding site conformations. The existence of 5 discrete $\Delta G_{LA}/\Delta G_{HA}$ ratios indicates that there are at least this many $AC_{LA}/AC_{HA}$ binding site structural pairs. Possible combinations that give rise to 5 $\eta$ classes are (number of $AC_{LA}$/number of $AO_{HA}$) 1/5, 2/3, 3/2, 4/2, and 5/1. For example, with the 1/5 combination a single $AC_{LA}$ binding site structure selects its target $AO_{HA}$ structure from among 5 alternatives.

The discrete distribution of $\eta$ classes suggests that the structural changes in the two stages of the induced fit also are discrete rather than continuous. The AChR allosteric transition combines elements of induced fit (use of binding energy to drive a local rearrangement) and conformational selection (constitutive activity and the adoption of pre-existing target structures) (*Changeux and Edelstein, 2005*).

The binding energy increase (in hold) occurs at the onset of the channel-opening gating isomerization (*Grosman et al., 2000*). The existence of multiple, distinct C versus O structural pairs at each binding site indicates that the end states in the simple activation scheme (*Figure 1B*) do not represent single stable structures but rather ensembles made up of receptors having at least five distinct binding site structural pairs. This raises the possibility that there could be five isomerization pathways that might culminate in five different structural perturbations of the gate region. Although there is

no evidence in AChRs of any agonist-dependent variations in output properties (conductance or ion selectivity), in other receptors functional output is agonist dependent (*Kenakin and Christopoulos, 2013*). Efficiency measurements of these receptors would test the possibility that $\eta$ classes are associated with the phenomenon of biased agonism (*Ehlert, 2018*).

Finally, regarding the agonist (*Figure 5*), there is a trend for those with cationic centers that occupy smaller volumes *in vacuo* to have greater $\eta$ (*Indurthi and Auerbach, 2021*). That is, agonists with larger volumes tend to use a greater fraction of their binding energy for catch. In AChRs and other receptors, the volume of the binding pocket appears to be smaller in O versus C (*Tripathy et al., 2019*), so it is also possible that unfavorable VdW interactions caused by pocket contraction guide the selection of (for example) the target O structure, to decrease $\eta$. The relationship between catch and hold energy change and the corresponding structural elements (agonist, protein, water) is complex and further investigations are needed.

## Extending η

The catch-and-hold LFER implied by an $\eta$ class indicates that structure (energy) changes in these two binding site rearrangements are related. Microscopic reversibility demands that if catch promotes hold, then hold promotes catch. Below, we present evidence suggesting that this bidirectional linkage is the tip of an iceberg, and that the entire AChR activation cascade - from an agonist at the encounter complex to water at the gate (^AC to AO) - is linked as an extended, multi-stage LFER (*Figure 2—figure supplement 1*).

There have been several proposals in AChRs regarding structural changes that link the neurotransmitter sites and the gate, but these are difficult to assess because they are without corroborating energy measurements. Likewise, $\eta$ measurements reveal a binding-gating energy link at the agonist site, but lack corroborating structural information. However, separate activation domains in AChR activation have been identified. For instance, $\eta$ calibrates the catch-hold connection at the agonist binding site, mutations decouple the ECD-TMD interface (*Cymes and Grosman, 2021*; *Shi et al., 2023*) and interactions between gate residues and pore water have been investigated (*Beckstein and Sansom, 2006*; *Rasaiah et al., 2008*; *Yazdani et al., 2020*). Analyses of the gating transition state suggest that there are 5 discrete domain rearrangements (*Figure 2—figure supplement 1*). In opening, hold is followed by movements of the ECD, TMD, gate region, and, finally, pore water and membrane lipids (*Gupta et al., 2017*).

Evidence for the extended LFER hypothesis is the experimental measurements of $k_{on}$ to C versus O. As mentioned above, when an unliganded AChR opens constitutively, $k_{on}$ increases to approach the diffusion limit, and also loses its agonist dependence. This indicates that with loop C intact, adopting the global O shape in the absence of agonists is sufficient to induce the local change in structure that removes the entry barrier, in hold. That a distant mutation (for example, of a gate residue) increases constitutive opening and also influences the ligand association rate constant is evidence that binding and gating are entangled.

This result, along with $\eta$ and $\Phi$ measurements, leads us to propose that everything in the protein's activation cascade, ^AC to AO, is energy-linked in a reversible, extended LFER. In this sequence, each individual domain energy change (rearrangement) influences those of its neighbors, bidirectionally. Usually, the first link in activation (after the encounter complex) is catch-and-hold, and the last is gate-and-water. However, in a LFER the consequence of a perturbation propagates both forward and backward, so a gate mutation would also perturb the agonist site and one at the ECD-TMD would induce rearrangements both there and at the gate. An extended LFER offers a way to transfer energy (change structure) over distance by using only local domain rearrangements. In AChRs, the allosteric transition appears to be a energy-coupled chain of domain rearrangements that crosses seamlessly the traditional divide between binding and gating. $\eta$ is at the core receptor operation insofar as it links the induced fit with the conformational gating cascade, $\Phi$ reveals the cascade's components (and the sequence of domain rearrangements), and $k_{on}$ to O reveals that a reverse cascade, gate→binding site, exists without agonists. Transit across the whole extended LFER is rapid (~5 μs; *Lape et al., 2008*), with each interemdiate states lasting on the order of 100 ns (*Gupta et al., 2017*).

There are many unanswered questions. We do not know the structural changes in the induced fit, or their energy consequences. Regarding agonists, we do not know the full spectrum of $\eta$ classes, or their structural correlates. Regarding mutations, we do not know the reasons specific agonist/side

chain combinations change $\eta$ or whether mutations of additional residues (including at a distance from the agonist site) can also do so. Regarding binding sites, we know little about the location of the encounter complex, why agonists are more efficient at the fetal AChR neurotransmitter site (*Nayak et al., 2019*), or $\eta$ values in other subtypes of nicotinic receptor or atl allosteric modulator binding sites. The extended LFER hypothesis needs to be tested, with both structure and energy changes identified and quantified. Regarding other receptors, there are few reports of the key experimental values ($K_{dO}$, $k_{on}$ to O, $\Phi$) so it is unknown whether $\eta$ classes, a staged induced fit, a conformational cascade, or an extended LFER apply generally. We anticipate that additional structure and energy measurements, in combination with computation, will reveal the molecular forces that underpin activation and lead to a deeper understanding of how ligands promote conformational change in allosteric proteins.

## Materials and methods
### Expression
Human embryonic kidney (HEK) 293 cells were maintained in Dulbecco's minimal essential medium (DMEM) supplemented with 10% FBS and 1% penicillin–streptomycin (pH 7.4). HEK cells obtained from ATCC (CRL-1573, lot no. 57925149) are authenticated using STR profiling and tested free of mycoplasma contamination. Mutations were incorporated into AChR subunits using the Quickchange II site directed mutagenesis kit (Agilent Technologies, CA) according to manufacturer's instructions. Sequence was verified by nucleotide sequencing (IDT DNA, I). AChRs were transiently expressed in HEK 293 cells by transfecting (CaPO$_4$ precipitation) (*Purohit et al., 2014*) mouse α1 (GFP encoded between M3-M4),β1,δ,ε subunits (3–5 μg total/ 35 mm culture dish) in a ratio of 2:1:1:1 for ~16 hrs. Most electrophysiological experiments were done 24–48 hr post-transfection.

### Electrophysiology
Single-channel currents were recorded in cell-attached patches (23 °C). The bath solution was (in mM) 142 KCl, 5.4 NaCl, 1.8 CaCl$_2$, 1.7 MgCl$_2$, 10 HEPES/KOH (pH 7.4). High extracellular [K$^+$] ensured that the membrane potential $V_m$ was ~0 mV. Patch pipettes were fabricated from borosilicate glass, coated with sylgard (Dow Corning, Midland, MI) to a resistance of ~10 MΩ when filled with pipette solution (Dulbecco's phosphate-buffered saline PBS) (in mM): 137 NaCl, 0.9 CaCl$_2$, 2.7 KCl, 1.5 KH$_2$PO$_4$, 0.5 MgCl$_2$, and 8.1 Na$_2$HPO$_4$ (pH 7.3/NaOH). Single channel currents were recorded using a PC505 amplifier (Warner instruments, Hamden, CT), low-pass filtered at 20 kHz and digitized at a sampling frequency of 50 kHz using a data acquisition board (SCB-68, National instruments, Austin, TX). For liganded activation experiments, agonists were added to the pipette solution at the desired concentrations. For unliganded activation experiments, we used pipettes and wires that were never exposed to agonists. To reduce the effect of channel block without affecting binding of agonist to the receptor, membrane potential ($V_m$) was held at +70 mV when agonists were used (*Jadey et al., 2011*).

### Current analysis
Analyses of the single-channel currents were performed by using QUB software (*Nicolai and Sachs, 2013*). Single-channel currents occur in clusters when the opening rate constant is significantly large. For analysis, we selected clusters of shut/open intervals that appeared (by eye) to be homogeneous, with regard to P$_o$. We limited the analysis to intracluster interval durations and thus excluded sojourns arising from desensitized states (shut intervals between clusters >20ms). The clusters were idealized into noise-free intervals after digitally filtering the data at 10–15 kHz (*Qin, 2004*). First, the idealized, intra-cluster intervals were fitted by a two-state model, C⇌O. Then, additional nonconducting and conducting states were added, one at a time, connected only to the first O state, until the log likelihood failed to improve by 10 units (*Qin et al., 1997*). Cluster P$_O$ at each agonist concentration was calculated from the time constants of the predominant components of the shut- ($\tau_s$) and open-time distributions ($\tau_o$): $\tau_o/(\tau_s + \tau_o)$. In this way, an equilibrium CRC was constructed as a plot of the absolute P$_O$ (not normalized) versus the agonist concentration (see *Figure 2*).

## Equilibrium constants

Two equilibrium dissociation constants comprise efficiency, $K_{dC}$ and $K_{dO}$ (*Figure 1*; *Equation 3*). These, and the fully-liganded gating constant $L_2$, were estimated in two ways, with both methods producing the same results.

In the primary approach, $K_{dC}$ and $K_{dO}$ were estimated from CRC parameters. The CRC was fitted by the Hill equation to estimate $P_O^{max}$ (the high concentration asymptote) and $EC_{50}$ (the agonist concentration that produces a half-maximum $P_O$). Equilibrium constants were calculated from the reaction scheme pertaining to the main activation pathway that assumes $L_0$ and $K_{dO}$ are negligible. Let $x=[A]/K_{dC}$. For a one-site receptor the scheme is $A+C \rightleftarrows AC \rightleftarrows AO$ and $P_O([A])= xL_1/(1+x + xL_1)$. For adult AChRs that have two equal and independent binding sites the scheme is $A+C \rightleftarrows AC \rightleftarrows A_2C \rightleftarrows A_2O$ (red, *Figure 1B*) and $P_O([A])= x^2L_2/(1+2x+x^2+x^2L_2)$. Relating the two site scheme to CRC parameters, $P_O^{max}$ is the infinite-concentration asymptote, $P_O^{min}$ is the zero-concentration asymptote and $EC_{50}$ is the concentration at which $P_O$ is half $P_O^{max}$,

$$L_2 = \frac{-P_O^{max}}{P_O^{max}-1} \tag{4a}$$

$$L_0 = \frac{-P_O^{min}}{P_O^{min}-1} \tag{4b}$$

$$K_{dC} = \frac{EC_{50}\left(L_2+1\right)}{1+\sqrt{L_2+2}} \tag{4c}$$

$$K_{dO} = \frac{K_{dC}}{\sqrt{\frac{L_2}{L_0}}} \tag{4d}$$

where $K_{dO}$ is solved by using *Equation 1*. Because $L_2=L_0c$ (*Equation 1*), any change in $L_0$ (see below) will change all three CRC parameters ($P_O^{max}$, $P_O^{min}$ and $EC_{50}$) even if the equilibrium dissociation constant ratio $K_{dC}/K_{dO}$ remains unchanged. Knowledge of, and ability to manipulate, $L_0$ was the key to measuring $\eta$.

## Voltage, $L_0$ and background mutations

To reduce channel block by the agonists, the membrane was depolarized to +70 mV. To compensate for changes in $\tau_s$ and $\tau_o$ caused by depolarization, we added the background mutation εS450W. This residue is far from the binding site (M4 transmembrane region of the ε subunit), has no effect on $K_{dC}$, and has equal but opposite effect on unliganded gating as does this extent of membrane depolarization (*Jadey et al., 2011*). $L_0$ is $7.4 \times 10^{-7}$ at $V_m$=-100 mV and reduced e-fold by a 60 mV depolarization (*Nayak et al., 2012*). Hence, we calculate that $L_0$ is $5.2 \times 10^{-7}$ at $V_m$=-70 mV as well as in our experiments at $V_m$ = +70 mV plus εS450W.

In some conditions, for instance low efficacy agonists (Dec, TriMA, and BzTEA) and aD200A, the wt opening rate constant was small and single-channel clusters were poorly defined. Accordingly, we added background mutations to facilitate $P_O$ measurements (*Tables 1 and 2*). These were εL269F (located in the M2 helix) and εE181W (located in strand β9) that increase the $L_0$ by 179- and 5.5-fold (1084-fold for the pair) without effecting $K_{dC}$. First, we obtained the apparent $L_2$ from the CRC $P_O^{max}$ (*Equation 4a*). Second, we divided this value by the fold increase in $L_0$ caused. By the background to obtain a corrected $L_2$. Finally, agonist $K_{dC}$ was estimated from $EC_{50}$ and the corrected $L_2$ (*Equation 4c*).

## $L_0$ for αD200A

In wt adult AChRs, $L_0$ is $5.2 \times 10^{-7}$ at $V_m$ = +70 mV (*Nayak et al., 2012*) and ofcourse is the same for all agonists. $L_0$ has been reported previously for the mutations αK145A (*Bruhova and Auerbach, 2017*) and αG153S (*Jadey et al., 2013*). To estimate this $L_0$ for αD200A, the pipette solution was free of any agonist and currents were measured at a membrane potential of −100 mV. The AChRs had added background mutations far from αD200 and each other (εL269F+εE181W+δV269A) that together increased unliganded activity substantially to allow cluster formation. Individually, these mutations increase $L_0$ by 179- (*Jha et al., 2009*), 5.5- (*Purohit et al., 2013*) and 250-fold (*Cymes et al., 2002*), respectively (*Figure 7—figure supplement 1*). Assuming no interaction (*Gupta et al., 2017*), the expected net increase in $L_0$ for this background combination is the product, ~$2.5 \times 10^5$. $L_0$ was measured experimentally using this background plus αD200A, from the durations of intra-cluster intervals (see above). The unliganded opening ($f_0$) and closing ($b_0$) rate constants were estimated from

the idealized interval durations by using a maximum-interval likelihood algorithm after imposing a dead time of 25 μs and $L_0$ was calculated from the ratio. Using a similar approach, $L_0$ was estimated previously for each mutation shown in *Figure 7*.

## Statistics

For single-channel CRCs, the midpoint and maximum ($EC_{50}$ and $P_O^{max}$) were estimated by fitting to monophasic Hill equation ($P_O = P_O^{max}/(1+(EC_{50}/[A])^{nH})$) using GraphPad Prism 6 (GraphPad). nH values contain information (*Qin, 2010*) but were not used because the number of binding sites was known a priori. A x-means cluster analysis algorithm (QUB online: qub.mandelics.com/online/xmeans.html) was used to define agonist (*Figure 4A*) and mutant (*Figure 7A*) classes, considering cluster must have at least two elements. Optimal clustering was determined based on the Sum Square Residual (SSR) and the corrected Akaike Information Criterion (AICc): for agonists SSR = $4.76 \times 10^{-3}$ and AICc = $-96.5$ (n=5 classes); for mutations SSR = $2.41 \times 10^{-2}$ and AICc=-396 (n=5 classes). Pearson's correlation test was performed to determine correlation significance between the two variables $logK_{dC}$ versus $logK_{dO}$ and $logL_2$ versus $log[1/K_{dC}]$. Since the goal was to measure the correlation rather than the magnitude difference between classes, Pearson's correlation was used instead of Cohen's. The P-value (two-tail) and $r^2$ value for that 3 agonist efficiency classes with >2 elements (*Figure 5A*) were 0.019 and 0.96 ($\eta$=0.56), <0.0001 and 0.99 ($\eta$=0.51) and <0.0001 and 0.99 ($\eta$=0.41). Significance for classes with <3 data points ($\eta$=0.31 and 0.46) could not be determined. The p-value (two-tail) and $r^2$ value for mutants (*Figure 7A*) were 0.0006 and 0.83($\eta$=0.56), <0.0001 and 0.98 ($\eta$=0.51), <0.0001 and 0.97 ($\eta$=0.45), <0.0001 and 0.93 ($\eta$=0.41), and <0.0001 and 0.98 ($\eta$=0.31). $\eta$ is the ratio of logarithms and as such is precise.

The position of the intermediate state $AC_{LA}$ in the catch-and-hold reaction sequence (1-$\eta$ ; *Figure 2*) is analogous to the position of the transition state ($\Phi$ or$\beta$) in a single-step reaction. With $\Phi$ the intermediate state in the LFER is an energy barrier, whereas with $\eta$ it is an energy well. $\Phi$ gives the fraction of the total energy change at ‡, and 1-$\eta$ gives the fraction of the total energy change at $AC_{LA}$.

## Agonists

Abbreviations: acetylcholine (ACh), trimethyl ammonium (TRiMA), tetramethyl ammonium, dimethylpyrrolidium (DMP), dimethylthiazolidinium (DMT), nornicotine, nicotinic (Nic), carbamylcholine (CCh), anabasine (Singh et al.), dimethylphenylpiperazinium (DMPP), benzyltrimethyl ammonium (BzTMA), choline (Cho), 3-hydroxypropyltrimethylammonium (3-OH), 4-hydroxybutyltrimethylammonium (4-OH), dimethylthiazolidinium (DMT), dimethylpyrrolidium (DMP), succinylcholine (SCh), decamethonium (Dec), epiboxidine (Ebx), epibatidine (Ebt), cytisine (Cyt), tetraethyl ammonium (TEA), tetramethyl phosphonium (TMP), varenicline (Var) and benzyltriethyl ammonium (BzTEA), Choline (Cho). Agonists were from Sigma (St. Louis, MO) except DMP, DMT, 3-OH and 4-OH that were synthesized as described previously (*Bruhova et al., 2013*).

## Acknowledgements

We thank Jan Jordan and Mary Teeling for technical assistance, and John Richard and Rob Phillips for helpful discussions.

## Additional information

### Funding

| Funder | Grant reference number | Author |
|---|---|---|
| NIH Blueprint for Neuroscience Research | NS-064969 | Anthony Auerbach |

The funders had no role in study design, data collection and interpretation, or the decision to submit the work for publication.

## Author contributions
Dinesh C Indurthi, Data curation, Software, Formal analysis, Investigation, Methodology; Anthony Auerbach, Conceptualization, Supervision, Funding acquisition, Writing - review and editing

## Author ORCIDs
Dinesh C Indurthi ![orcid] http://orcid.org/0000-0001-8837-5883
Anthony Auerbach ![orcid] http://orcid.org/0000-0003-4151-860X

## Decision letter and Author response
Decision letter https://doi.org/10.7554/eLife.86496.sa1
Author response https://doi.org/10.7554/eLife.86496.sa2

---

# Additional files

## Supplementary files
• MDAR checklist

## Data availability
The source data is available in the Tables and table source data files and referred to in the legend of the associated figure.

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
