## [Editor Report]

This valuable work investigates the fundamental concept of how the energy of agonist binding is converted into the energy of the conformational change that opens the pore of a ligand-gated ion channel. The conclusions are based on analysis of solid data in terms of a mechanistic model. The findings will be interesting to biophysicists working on ligand-gated ion channels and, more generally, to enzymologists focused on allosteric enzyme regulation.

---

## [Decision Letter]

**Decision letter after peer review:**

Thank you for submitting your article "Agonist efficiency links binding and gating in a nicotinic receptor" for consideration by *eLife*. Your article has been reviewed by 3 peer reviewers, including László Csánady as the Reviewing Editor and Reviewer #1, and the evaluation has been overseen by Kenton Swartz as the Senior Editor. The following individual involved in review of your submission has agreed to reveal their identity: Andrew J R Plested (Reviewer #2).

Essential revisions:

1. Generality

The authors repeatedly state that the concept of efficiency is generally applicable, but this is not demonstrated. Moreover, "efficiency" (eta) is defined in multiple different ways. E.g.: (i) In the Table 1 legend: "eta=1-logKdC/logKdO". That definition is certainly applicable to any receptor, as KdC and KdO values always exist. (ii) On line 325: "efficiency is the correlation between affinity and efficacy. It is a universal parameter that applies to every agonist of every receptor." Not only is the latter definition not shown to be universally applicable, but it is not even meaningful: the data presented here shows that affinity and efficacy are not "correlated", as their ratio can scatter between <0.3 and >0.6 (Figure 7B). Thus, the authors should more exactly define what they mean by "the concept of efficiency", and clearly spell out what is generally true vs. what is shown to be true specifically for the nAChR and only assumed to be true in general.

2. Definition of eta-classes

One concern is related to the statistical treatment of the measured and calculated values.

No statistical evaluation (SD, confidence intervals, p-values) is given for the calculated metrics- KdC, KdO, c, eta. Surely, the errors of the values used in calculations should propagate to the resulting calculus results. Uncertainty of these calculated metrics matter because based on them principal conclusions are made. It is not clear at all if variability of efficiencies would affect grouping of them as shown in Figure 7. Presenting the points with the error bars would give the reader a test of how confident one could be in groups of efficiencies.

The reviewers are concerned about the clustering of the data set into exactly five classes. First, the statistical treatment of the measured and calculated values is a major concern. The symbols in the efficiency plots (Figures 5a-b, 7a-b) lack error estimates: no statistical evaluation (SD, confidence intervals, p-values) is given for the calculated metrics KdC, KdO, DDG, or eta. Surely, the errors of the values used in the calculations should propagate to the resulting calculated variables. Uncertainty of these calculated metrics matter because based on them principal conclusions are made. It is not clear if the variability of efficiencies would affect grouping of them as shown in Figure 7. Presenting the points with the error bars would give the reader a taste of how confident one can be in the grouping of efficiencies. Second, even without error estimates, if the dots in Figure 5b were not color coded they would not appear to be segregated into exactly five clusters each of which fall on distinct straight lines. Third, there is no theoretical foundation for the existence of a discrete number of eta-classes as opposed to a continuum of possible eta values: the catch&hold LFER model presented in Figure 2 does not predict this.

3. Uniqueness of the catch&hold LFER model

The authors do not discuss the uniqueness of the proposed catch&hold LFER model (Figure 2) used for data interpretation. It seems that the existence of eta-classes might be explained just as well by an alternative model which assumes a single gating mechanism for the receptor, but distinct patterns of ligand-protein interactions for the different agonists (see Review Figure linked to the decision letter). This fact should be acknowledged, or if data exist to differentiate between these possibilities they should be presented and discussed.

4. Differentiating new from old data

The authors should clearly indicate what new data and what old data are included in each figure so the readers can judge the claimed advance. Replotting old data in a new way is acceptable, but throughout the manuscript it should be clearly spelt out what are actual new data and what are published data that have been replotted.

5. Clarity of presentation

All three reviewers have pointed out numerous places where the presentation is unclear. The paper would benefit from a major re-writing along the following specific guidelines: the authors should (i) provide only a single definition for a given variable (e.g., eta), (ii) avoid introduction of unnecessary parameters (e.g., kappa), (iii) expose the theory in a single block in the main text (e.g., in Intro or beginning of Results), and (iv) address specific comments regarding lack of clarity raised in all three reviews.

Finally, the reviewers feel that the attached movie does not provide additional information relative to that presented in the paper, and it is also much longer than a typical concise summary video. Thus, linking it to the manuscript seems unnecessary.

*Reviewer #1 (Recommendations for the authors):*

1. What exactly is plotted in Figure 7b? What does the x axis represent?

2. Normalization (line 577): "The free energies DeltaG(LA) and DeltaG(HA) (Figure 1B), proportional to logKdC and logKdO, are each a sum of a ligand-protein binding energy and a chemical potential that incorporates the energy consequence of removing the ligand from solution".

Shouldn't DeltaG(LA) and DeltaG(HA) actually represent standard free energies of binding, which are devoid of the log-concentration term? (The expression -RT*lnKeq gives standard free energy change.)

*Reviewer #2 (Recommendations for the authors):*

There are many interesting points made – it is not necessarily new to think of agonists like mutations but this paper underlines the similarity.

1. One criticism I have is exemplified by statements like the following:

line 214 "Every agonist of every receptor has an affinity, efficacy and efficiency."

line 39 "Recently, a third universal agonist property, efficiency (; eta), was introduced as the correlation between affinity and efficacy"

I know that these assertions make sense to the authors. On paper, they seem reasonable. But they are not tested. There is no general demonstration. There is no information that they apply to any other receptor. Of course, the senior author's lab has developed tools over decades to test these ideas in muscle nicotinic receptors. And this is a totally valid approach. But we have no information about whether this concept is universal or special to the nAChR.

I am not expecting the authors to go back and prove that this idea is general for this paper. But the statements could be toned down because they lack evidence.

As a side note, how would this apply for example to g-protein coupled receptors? Is it generally applicable to receptors with co-ligands, co-factors? Is it applicable for other types of receptors? Or is it limited to (in as much as I mean, only useful for) ion channel receptors with nice gating properties. I think the reader deserves an honest assessment. Can it be reasonably concluded that it is too difficult to assess these properties in other systems?

A related problem is that, as was discussed in the past, the citations are overwhelmingly papers from the senior author's own lab. I did not count the percentage. It's close to half, I think. I know that there is some justification for this, but it speaks against the generality of the work. It suggests that this concept is in fact rather narrow.

2. Overall I am not sure why the treatment oscillates in the paper between kappa and eta. It took me quite a long time reading to recognise that kappa = 1 – eta and this makes understanding this technical work a tick harder.

I would stick to one and only mention the other in exceptional circumstances.

3. line 202 We measured kappa and eta in AChRs having wild-type (wt) binding sites for 5 previously unstudied agonists

I think this is the wrong way to put it. Decamethonium has been studied a great deal (e.g. https://www.pnas.org/doi/pdf/10.1073/pnas.75.6.2994), as has SCh. Do you mean, comparatively understudied by single channel recording? But there were already a few papers…

4. It would be nice in Figure 3 to note the voltage, and the mutants that are employed. It's not the same mutant for each curve. I think these records should have been recorded at +70 mV (as described in the methods, to avoid block) but the openings are downwards. Is this right? Inward currents that were flipped? Sorry, I think I have something backwards but I don't understand.

5. line 341 The main results are as follows. (1) Empirically, efficiency is the log-log correlation between affinity and efficacy. It is a universal parameter that applies to every agonist of every receptor.

Is that a result that is shown in this paper? Same goes for result (2) in this list, which seems not to be a result from this paper but rather a postulate taken on the basis of existing results.

6. line 317 "Figure 7A shows a kappa-plot for all mutations that have been measured so far."

I'm confused. This panel includes results from previous work. There are >50 measurements of Kappa in Figure 7. But there are not 50 measurements of mutants in Table 2-supplement 1 and anyway nearly half of those in the table are repeats on different agonists. Can you be explicit about the mutants that are in Figure 7? The strength of the conclusion that the groups are the same for binding site mutations and for agonists rests on a more precise report of what is plotted here.

*Reviewer #3 (Recommendations for the authors):*

First of all, I would like to compliment the Authors for producing an impressive set of experimental data. The amount of single channel recordings produced is truly formidable.

One concern is related to the statistical treatment of the measured and calculated values. Only the EC50 and Po estimates have standard errors of the mean presented in Table 1. However, no statistical evaluation (SD, confidence intervals, p-values) is given for the calculated metrics- KdC, KdO, c, nu. Surely, the errors of the values used in calculations should propagate to the resulting calculus results. Uncertainty of these calculated metrics matter because based on them principal conclusions are made.

It is not clear at all if the variability of efficiencies would affect the grouping of them as shown in Figure 7. Presenting the points with the error bars would give the reader a test of how confident one could be in groups of efficiencies.

One example of the problems related to the statistical treatment is the following. A dissociation rate constant, koff, value used in calculations is 15,000 1/s. The Authors accept that it is approximate (lines 556 and 557), however, no standard deviation or confidence intervals are given. Surely, the error of this value should propagate into all calculated values where k_off_ is used. On the other hand, k_off_ was estimated and is approximately the same for few agonists tested in Jadey & Auerbach 2012 but is it the same for many more agonists in this study?

Another concern about this manuscript is that it is written in a very confused and convoluted style. The manuscript is very difficult to follow. The entire section of 'Theory' in the results and a corresponding section in the Methods should be united (for fluency of reading) and use more straightforward and exact language. Indeed, I might have misunderstood some of the arguments the Authors make and I'm not sure I fully understood the theoretical background.

Here are a few comments in order of appearance I wrote down before giving up. It is in no way an exhaustive list.

Line 36: It is not clear what 'Receptor theory' refers to.

Lines 38-39: efficiency- universal agonist property. Is it really universal? There is no supporting evidence for that beyond the AChR receptor analysed in this particular framework advocated by the authors.

Lines 46-47: activation to a first approximation… What is activation to a second and further approximations?

Lines 49-50: 'regions change structure and function conjointly'. It is not a very clear expression. Do authors say that conformation of both- binding site and gate- change concertedly/instantaneously? Does the binding site change conformation as well when spontaneous opening happens?

Line 53-55: sentence is very unclear. Is reference to Figure 1 supposed to show an increase in Po and membrane current? What exactly does 'membrane current' in this sentence mean? Is the reference to Figure 1 supposed to be to Figure 1A to demonstrate 'favorable binding free energy to O…'? Yet it is not clear from Figure 1A what it is.

Lines 55-57: very confusing sentence. 'Asymptotes of CRC' come out of the blue and a nonexpert reader could be left very disconcerted.

Lines 58-59: Without finishing proper introduction (Introduction will continue in a paragraph below) the Authors declare what their intention is. The intention 'is to estimate free energy changes associated with each step in the process'. At this stage we have no idea what 'the process' is and even less idea about the steps in 'the process'. The rest of this paragraph does not get much clearer.

Line 64: 'weak/strong ratio': ambiguous.

Lines 108-109: 'two apparently independent steps'- do the Authors mean events? Surely, there are more than two steps in the scheme in Figure 1B.

Figure 1, text and several equations: it is not clear why L is used for efficacy in this manuscript despite the fact that letter E is used in the field and, indeed, in an earlier paper by the same authors.

Words 'bind' and 'gate' seem to be used in a very liberal way.

---

## [Author Response]

New statistics Ln 258 (Figure 4A), 291-297 (Figure 4B), 345-349 (Figure 7A), 353-355 (Figure 7B). We used the x-means algorithm and AICc metric to select the optimal number as n=5 classes (Ln 666).Figures4B and 7B. we note that the sd of each point is smaller than the symbol.Ln 429. Number of classes

Ln 241 h is not sensitive to errors in EC_50_ and P_O_^max^. h is precise (a ratio of logs). Agonist classes that are poorly defined in Figure 4B are supported by the mutational classes in Figure 7.

The reviewers are concerned about the clustering of the data set into exactly five classes. First, the statistical treatment of the measured and calculated values is a major concern. The symbols in the efficiency plots (Figures 5a-b, 7a-b) lack error estimates: no statistical evaluation (SD, confidence intervals, p-values) is given for the calculated metrics KdC, KdO, DDG, or eta. Surely, the errors of the values used in the calculations should propagate to the resulting calculated variables. Uncertainty of these calculated metrics matter because based on them principal conclusions are made. It is not clear if the variability of efficiencies would affect grouping of them as shown in Figure 7. Presenting the points with the error bars would give the reader a taste of how confident one can be in the grouping of efficiencies. Second, even without error estimates, if the dots in Figure 5b were not color coded they would not appear to be segregated into exactly five clusters each of which fall on distinct straight lines. Third, there is no theoretical foundation for the existence of a discrete number of eta-classes as opposed to a continuum of possible eta values: the catch&hold LFER model presented in Figure 2 does not predict this.

Correct Ln 148 403 421.

3. Uniqueness of the catch&hold LFER modelThe authors do not discuss the uniqueness of the proposed catch&hold LFER model (Figure 2) used for data interpretation.

Ln 420 (see below for generality of induced fit/clamshell). In AChRs, the catch-hold LFER is correct.

It seems that the existence of eta-classes might be explained just as well by an alternative model which assumes a single gating mechanism for the receptor, but distinct patterns of ligand-protein interactions for the different agonists (see Review Figure linked to the decision letter). This fact should be acknowledged, or if data exist to differentiate between these possibilities they should be presented and discussed.

Ln 141-145 We repeat the distinction between efficacy (λ) and efficiency (eta). The cartoons only depict ‘hold’ (in Figure 2, AC_LA_=AC_HA_). We are not sure what a “single gating mechanism” means (see below). We think that the contact ratio idea is too simplistic:

AChR agonists are small and the physical basis of a ‘contact’ is not clear.It is extremely unlikely that every contact has an identical deltaG.Ln 170-190 K_d_ is set by ligand/protein/water rearrangements (induced fits) rather than only ligand-protein contacts.The cartoons do not incorporate catch (the slow k_on_ to C, SI Figure 2 legend).Ln 444-449 Our experimental focus in this paper is energy and without data we limit our discussion of structure to what we learn from h. We need to have structures of ^A^C, AC_LA_ and AC_HA_ w/ different agonists.

4. Differentiating new from old dataThe authors should clearly indicate what new data and what old data are included in each figure so the readers can judge the claimed advance. Replotting old data in a new way is acceptable, but throughout the manuscript it should be clearly spelt out what are actual new data and what are published data that have been replotted.

Ln 86-88, 229 (agonists), 304-306 (mutations), Table titles (Main and SI).

5. Clarity of presentationAll three reviewers have pointed out numerous places where the presentation is unclear. The paper would benefit from a major re-writing along the following specific guidelines: the authors should (i) provide only a single definition for a given variable (e.g., eta), (ii) avoid introduction of unnecessary parameters (e.g., kappa),

See above; eta defined in Eq 2; kappa is gone.

(iii) expose the theory in a single block in the main text (e.g., in Intro or beginning of Results), and (iv) address specific comments regarding lack of clarity raised in all three reviews.Finally, the reviewers feel that the attached movie does not provide additional information relative to that presented in the paper, and it is also much longer than a typical concise summary video. Thus, linking it to the manuscript seems unnecessary.

Ln 110-145 single block – affinity vs efficacy vs efficiency.

Ln 170-182 “catch” and “hold” replace the confusing “bind” and “gate”.

Reviewer #1 (Recommendations for the authors):1. What exactly is plotted in Figure 7b? What does the x axis represent?

Figure 7B has been changed to DG_LA_ vs DG_HA_. The DDG plot is now in the SI Figure 7.

2. Normalization (line 577): "The free energies DeltaG(LA) and DeltaG(HA) (Figure 1B), proportional to logKdC and logKdO, are each a sum of a ligand-protein binding energy and a chemical potential that incorporates the energy consequence of removing the ligand from solution".Shouldn't DeltaG(LA) and DeltaG(HA) actually represent standard free energies of binding, which are devoid of the log-concentration term? (The expression -RT*lnKeq gives standard free energy change.)

Ln 110-116 DG defined and examples given.

Ln 191-203, SI Figure 7. Consideration of the chemical potential (binding entropy)

Reviewer #2 (Recommendations for the authors):There are many interesting points made – it is not necessarily new to think of agonists like mutations but this paper underlines the similarity.1. One criticism I have is exemplified by statements like the following:line 214 "Every agonist of every receptor has an affinity, efficacy and efficiency."

deleted (replaced by Ln 146-151).

line 39 "Recently, a third universal agonist property, efficiency (eta), was introduced as the correlation between affinity and efficacy"

This statement is true.

I know that these assertions make sense to the authors. On paper, they seem reasonable. But they are not tested. There is no general demonstration. There is no information that they apply to any other receptor. Of course, the senior author's lab has developed tools over decades to test these ideas in muscle nicotinic receptors. And this is a totally valid approach. But we have no information about whether this concept is universal or special to the nAChR.

See above. We have toned down the language re: universality.

Ln 146 Eq. 2 If K_dC_ and K_dO_ are universal then by definition so, too, is h. If A and B are universal, then so is A/B.

Ln 556-558 What needs to be tested by experiment are whether efficiency *classes* and our physiochemicial interpretation of h are general.

Ln 402-405 KdC, KdO and k_on_ to O have not been measured in other receptors.

Ln 424, 428 Structures of the key intermediate states have not been identified.

Ln 622 L_0_ (also mostly unknown in other receptors) is the key to measuring eta.

I am not expecting the authors to go back and prove that this idea is general for this paper.

We think our job is to drill deeper into the AChR rather than dig wider into other receptors. We hope others will measure efficiency in their receptor.

But the statements could be toned down because they lack evidence.

Language has been toned down.

As a side note, how would this apply for example to g-protein coupled receptors? Is it generally applicable to receptors with co-ligands, co-factors? Is it applicable for other types of receptors? Or is it limited to (in as much as I mean, only useful for) ion channel receptors with nice gating properties. I think the reader deserves an honest assessment. Can it be reasonably concluded that it is too difficult to assess these properties in other systems?

Ln 419 We mention a GPCR. Insofar as Figure 1 is general, then so, too is h. Any bistable protein activated by a difference in stimulus energy – GPCR, Hb, GroEL, , LGICs w/ cofactors, VGICs… – has an associated efficiency (the ratio of stimulus energies to off/on conformations). However, the existence of efficiency classes and generality of the induced fit mechanism need to be tested by experiment. Our hypothesis is that these, too, could be general (see above).

Ln 415-423 slow k_on_ to C and h plots from published results (Nayak et al. 2019) hint that h classes exist in other receptors, including a GPCR. There are caveats regarding testing for h classes in other receptors:

Measuring the energies accurately by experiment can be problematic. Affinities switch readily and it can be hard to measure K_dC_ and K_dO_ cleanly. Also, state models for some receptors can be complex. The place to start is L_0_.There could be additional binding energy changes buried in the data. However, downstream energy changes that **do not** involve a ligand energy change show up as an offset to the y axis of the h plot and do not affect the h estimate. Post-hold events (post-gating in GPCRs) that are the same energetically for all agonists simply translate the plot up/ down but do not change the slope.If there is an external energy source, or if occupancy of the site by an agonist has a global effect on the protein other than gating, then Figure 1 is void. The key assumptions in Figure 1 (the thermodynamic cycle) are (1) that differential ligand binding energies simply add to an intrinsic gating energy that is the same with or without the ligand, and (2) there is no significant external energy. In AChRs, both have been proved experimentally (Nayak 2017). Moreover, mutant analyses in AChRs show that energy changes (including from agonist binding) are independent and local (additive when separated by >12 A) (Gupta 2017).

We don’t discuss these caveats because we think they would be a distraction.

A related problem is that, as was discussed in the past, the citations are overwhelmingly papers from the senior author's own lab. I did not count the percentage. It's close to half, I think. I know that there is some justification for this, but it speaks against the generality of the work. It suggests that this concept is in fact rather narrow.

We do not agree. There can be other reasons for few external citations.

2. Overall I am not sure why the treatment oscillates in the paper between kappa and eta. It took me quite a long time reading to recognise that kappa = 1 – eta and this makes understanding this technical work a tick harder.I would stick to one and only mention the other in exceptional circumstances.

kappa is gone.

3. line 202 We measured kappa and eta in AChRs having wild-type (wt) binding sites for 5 previously unstudied agonists.I think this is the wrong way to put it. Decamethonium has been studied a great deal (e.g. https://www.pnas.org/doi/pdf/10.1073/pnas.75.6.2994), as has SCh. Do you mean, comparatively understudied by single channel recording? But there were already a few papers…

Ln 235 wording changed.

4. It would be nice in Figure 3 to note the voltage, and the mutants that are employed. It's not the same mutant for each curve. I think these records should have been recorded at +70 mV (as described in the methods, to avoid block) but the openings are downwards. Is this right? Inward currents that were flipped? Sorry, I think I have something backwards but I don't understand.

Currents are now inverted; V_m_ and the background mutations are noted in the legends.

Ln 625-632 The background is discussed in the Methods The mutation eS450W compensates exactly for the effect of depolarization on gating without altering binding. It was discovered long ago (Jadey and Auerbach, 2011) has been referenced in perhaps a dozen papers. With this mutation, the duration of single-channel outward current intervals at +70 mV (no channel block) are the same as inward intervals at -100 mV, which is to say long and therefore easy to measure.

5. line 341 The main results are as follows.

The list has been deleted from the Discussion.

1) Empirically, efficiency is the log-log correlation between affinity and efficacy. It is a universal parameter that applies to every agonist of every receptor.

See above.

Is that a result that is shown in this paper? Same goes for result (2) in this list, which seems not to be a result from this paper but rather a postulate taken on the basis of existing results.

See above.

6. line 317 "Figure 7A shows a kappa-plot for all mutations that have been measured so far." I'm confused. This panel includes results from previous work. There are >50 measurements of Kappa in Figure 7. But there are not 50 measurements of mutants in Table 2-supplement 1 and anyway nearly half of those in the table are repeats on different agonists. Can you be explicit about the mutants that are in Figure 7?

All are given in Table 2 and SI Table2.

The strength of the conclusion that the groups are the same for binding site mutations and for agonists rests on a more precise report of what is plotted here.Reviewer #3 (Recommendations for the authors):First of all, I would like to compliment the Authors for producing an impressive set of experimental data. The amount of single channel recordings produced is truly formidable.One concern is related to the statistical treatment of the measured and calculated values. Only the EC50 and Po estimates have standard errors of the mean presented in Table 1. However, no statistical evaluation (SD, confidence intervals, p-values) is given for the calculated metrics- KdC, KdO, c, nu. Surely, the errors of the values used in calculations should propagate to the resulting calculus results. Uncertainty of these calculated metrics matter because based on them principal conclusions are made.It is not clear at all if the variability of efficiencies would affect the grouping of them as shown in Figure 7. Presenting the points with the error bars would give the reader a test of how confident one could be in groups of efficiencies.

See above. The error bars in Figure 4B and 7B are all smaller than the symbol.

One example of the problems related to the statistical treatment is the following. A dissociation rate constant, koff, value used in calculations is 15,000 1/s. The Authors accept that it is approximate (lines 556 and 557), however, no standard deviation or confidence intervals are given. Surely, the error of this value should propagate into all calculated values where k_off_ is used. On the other hand, k_off_ was estimated and is approximately the same for few agonists tested in Jadey & Auerbach 2012 but is it the same for many more agonists in this study?

This discussion has been moved to the SI Figure 2. We calculated eta from equilibrium constants estimated from a CRC (we did not measure k_off_). This is a thought experiment and meant only to emphasize bind-gate entanglement. We are not suggesting that k_on_ measurements should replace actual measurements of CRCs and tau. However, we think it’s worth noting that an association rate constant can predict gating responses.

Another concern about this manuscript is that it is written in a very confused and convoluted style. The manuscript is very difficult to follow. The entire section of 'Theory' in the results and a corresponding section in the Methods should be united (for fluency of reading) and use more straightforward and exact language. Indeed, I might have misunderstood some of the arguments the Authors make and I'm not sure I fully understood the theoretical background.Here are a few comments in order of appearance I wrote down before giving up. It is in no way an exhaustive list.

We hope that our changes (the earlier introduction of catch-hold and induced fits) alleviate some of the confusion.

Line 36: It is not clear what 'Receptor theory' refers to.

Gone.

Lines 38-39: efficiency- universal agonist property. Is it really universal? There is no supporting evidence for that beyond the AChR receptor analysed in this particular framework advocated by the authors.

See above.

Lines 46-47: activation to a first approximation… What is activation to a second and further approximations?

Ln 479-485 In the original we used “first approximation” because the results suggest there are 5 C/O structural pairs and many intermediate transitions. Perhaps some confusion is caused by taking the simple state model (Figure 1B) and single-channel current traces (on-off) too literally. Rather, schemes and patch clamp recordings are approximations that do not reflect the reality of proteins. In a scheme, each capital letter (state) represents an ensemble and each arrow (transition) contains intermediates, both of which are invisible in recordings. Figure 2A and SI Figure 2 expand the standard scheme to allow us to interpret experimental measurements that are aggregates (DG_LA_, DG_HA_, k_on_, F). One needs to imagine 5 stable C/O pairs and jittering intermediate transitions between states having ns-us lifetimes. These can be inferred from energy measurements even if the corresponding structures have not yet been identified. The classic example of a useful, if undetected, intermediate state is Michaelis-Menton.

Lines 49-50: 'regions change structure and function conjointly'. It is not a very clear expression.

We don’t think this phrase is vague, but “conjointly” is redundant and therefore has been deleted.

Do authors say that conformation of both- binding site and gate- change concertedly/instantaneously?

The definition of these words depends on the time scale (see above). Neither applies on the ps time scale, but both probably apply on the ms scale. We think about the AChR in ns-ms range and so avoid these words.

Does the binding site change conformation as well when spontaneous opening happens?

Ln 475-478. We don’t know.

Line 53-55: sentence is very unclear. Is reference to Figure 1 supposed to show an increase in Po and membrane current? What exactly does 'membrane current' in this sentence mean?

Changed. Single channel P_O_ is proportional to membrane current (whole cell current = nP_O_i, where i is the single channel current and n is the number of channels and equal to 1 in our experiments). This relationship is so standard we don’t think it’s worth mentioning.

Is the reference to Figure 1 supposed to be to Figure 1A to demonstrate 'favorable binding free energy to O…'? Yet it is not clear from Figure 1A what it is.

Figure 1 shows the same thing – the thermodynamic cycle – in 2 different ways. A, The blue lines in the landscape represent binding energies, stronger (longer) to O versus C. B, The cycle is easier to see when written as a reaction scheme. We don’t see how we can make be any clearer.

Lines 55-57: very confusing sentence. 'Asymptotes of CRC' come out of the blue and a nonexpert reader could be left very disconcerted.

Ln 59 Changed.

Lines 58-59: Without finishing proper introduction (Introduction will continue in a paragraph below) the Authors declare what their intention is. The intention 'is to estimate free energy changes associated with each step in the process'. At this stage we have no idea what 'the process' is and even less idea about the steps in 'the process'. The rest of this paragraph does not get much clearer.

Ln 56-57 wording changed.

Line 64: 'weak/strong ratio': ambiguous.

Ln 63 changed.

Lines 108-109: 'two apparently independent steps'- do the Authors mean events? Surely, there are more than two steps in the scheme in Figure 1B.

Ln 105-109 changed.

Figure 1, text and several equations: it is not clear why L is used for efficacy in this manuscript despite the fact that letter E is used in the field and, indeed, in an earlier paper by the same authors.

Some people complained about our previous use of E (some use it for energy). L is the standard for the gating equilibrium constant in VGICs (including BK channels that are activated by ligands). In this paper we are consistent do not think that the change will generate significant confusion, especially because we mainly consider K_d_s.

Words 'bind' and 'gate' seem to be used in a very liberal way.

Figure 1B These words are defined explicitly in the reaction scheme.

Ln 170-182 Figure 2 and SI Figure 2. We now use ‘catch’ and ‘hold’, the microscopic subevents inside bind and gate that are germane to eta. ‘catch’ and ‘hold’ may not be as memorable as ‘flip’ (perhaps because they are more complex/abstract), but they are informative and useful.